# Assessing Potential Indicators of Aerosol Wet Scavenging During Long-Range Transport

Miguel Ricardo A. Hilario[1], Avelino F. Arellano[1], Ali Behrangi[1,2], Ewan C. Crosbie[3,4], Joshua P. DiGangi[3], Glenn S. Diskin[3], Michael A. Shook[3], Luke D. Ziemba[3], and Armin Sorooshian[1,5]

[1] Department of Hydrology and Atmospheric Sciences, University of Arizona, Tucson, AZ, USA
[2] Department of Geosciences, University of Arizona, Tucson, AZ, USA
[3] NASA Langley Research Center, Hampton, VA, USA
[4] Science Systems and Applications, Inc., Hampton, VA, USA
[5] Department of Chemical and Environmental Engineering, University of Arizona, Tucson, AZ, USA

*Correspondence to*: Armin Sorooshian (armin@arizona.edu)

**Abstract.** As one of the dominant sinks of aerosol particles, wet scavenging greatly influences aerosol lifetime and interactions with clouds, precipitation, and radiation. However, wet scavenging remains highly uncertain in models, hindering accurate predictions of aerosol spatiotemporal distributions and downstream interactions. In this study, we present a flexible, computationally inexpensive method to identify meteorological variables relevant for estimating wet scavenging using a
15 combination of aircraft, satellite, and reanalysis data augmented by trajectory modelling to account for air mass history. We assess the capabilities of an array of meteorological variables to predict the transport efficiency of black carbon ($TE_{BC}$) using a combination of nonlinear regression, curve-fitting, and k-fold cross-validation. We find that accumulated precipitation along trajectories (APT) – treated as a wet scavenging indicator across multiple studies – does poorly when predicting $TE_{BC}$. Among different precipitation characteristics (amount, frequency, intensity), precipitation intensity was the most effective at estimating
$TE_{BC}$ but required longer trajectories ($> 48$ h) and including only intensely precipitating grid cells. This points to the contribution of intense precipitation towards aerosol scavenging and the importance of accounting for air mass history. Predictors that were most able to predict $TE_{BC}$ were related to the distribution of relative humidity (RH) or the frequency of humid conditions along trajectories, suggesting that RH is a more robust way to estimate $TE_{BC}$ than APT. We recommend the following alternatives to APT when estimating aerosol scavenging: (1) the 90th percentile of RH along trajectories, (2) the fraction of hours along trajectories
with either water vapor mixing ratios $> 15$ g kg$^{-1}$ or RH $> 95\%$, (3) precipitation intensity along trajectories at least 48 hours along and filtered for grid cells with precipitation $> 0.2$ mm h$^{-1}$. Future scavenging parametrizations should consider these meteorological variables along air mass histories. This method can be repeated for different regions to identify region-specific factors influencing wet scavenging.

# 1 Introduction

Although wet scavenging is one of the dominant removal mechanisms for atmospheric aerosol particles (Seinfeld and Pandis, 2016; Textor et al., 2006), it remains a large source of uncertainty in global-scale models (Watson-Parris et al., 2019; Liu and Matsui, 2021; Moteki et al., 2019; Hodzic et al., 2016). This uncertainty hampers the ability of global-scale models to capture the lifecycle (i.e., sources, transformations, and sinks) (Hou et al., 2018), spatial extent (Moteki et al., 2019), and vertical profile (Watson-Parris et al., 2019; Liu and Matsui, 2021; Frey et al., 2021; Kipling et al., 2016) of aerosol particles. Inaccurate representations of these aerosol features contribute to uncertainties in estimates of aerosol radiative effects (Samset et al., 2013; Marinescu et al., 2017) and aerosol loadings over climate-sensitive regions (Liu and Matsui, 2021; Mahmood et al., 2016; Shen et al., 2017), with further implications for the remote sensing of aerosol abundance downwind of precipitating or cloudy areas. Advancing knowledge of wet scavenging processes can help reduce the largest uncertainty in human forcing of the climate system, which involves aerosol-cloud interactions (e.g., Bellouin et al., 2020).

Wet scavenging occurs either below- or in-cloud. Below-cloud scavenging occurs when aerosol particles are collected by precipitation (Croft et al., 2009) and is most important between the surface and 1 km above ground level (AGL) (Grythe et al., 2017). The efficiency of below-cloud scavenging depends on raindrop size distributions (Wang et al., 2010), aerosol composition (Lu and Fung, 2018; Grythe et al., 2017), the amount of in-cloud condensed water (Luo et al., 2019) as well as precipitation characteristics (i.e., frequency, intensity, amount, and type). To calculate the fraction of aerosol scavenged below-cloud, models typically rely on an empirically-derived below-cloud scavenging coefficient, which is a function of aerosol size (Feng, 2007; Croft et al., 2009) and composition (Lin et al., 2021). Semi-empirical model parametrizations of below-cloud scavenging have been shown to improve simulated surface concentrations (Luo et al., 2019); however, agreement between models and observations is highly sensitive to the specific below-cloud scavenging scheme used (Lu and Fung, 2018). Below-cloud scavenging rates in models also remain significantly underestimated compared to observations (Kim et al., 2021; Ryu and Min, 2022; Xu et al., 2019).

Wang et al. (2010) determined the below-cloud scavenging coefficient is influenced by (1) raindrop-particle collection efficiency, (2) raindrop size distribution, and (3) raindrop terminal velocity. These factors were associated with differences in particle concentrations by a factor of 2 for sub-10 nm particles and a factor of >10 for particles larger than 3 µm; however, their combined uncertainty was insufficient to explain the discrepancy between theoretical and field measurements of the below-cloud scavenging coefficient. Wang et al. (2011) demonstrated that this discrepancy can be largely explained by the vertical turbulence as it determines which particles are subjected to impaction scavenging. This impact was most pronounced for submicron particles under weak precipitation intensities.

Given these uncertainties, Wang et al. (2014a) developed a new semi-empirical, size-resolved parametrization based on an percentile-logarithmic power-law relationship between the below-cloud scavenging coefficient and particle size that is applicable to both rain and snow across different particle sizes and precipitation intensities. Based on the size-resolved parametrization of Wang et al. (2014a), a bulk or modal parametrization for fine ($PM_{2.5}$), coarse ($PM_{2.5-10}$), and giant particles ($PM_{10+}$) was presented by Wang et al. (2014b).

In-cloud scavenging occurs via nucleation (i.e., activation of aerosol particles into cloud droplets; Jensen and Charlson, 1984) or impaction (i.e., collision of interstitial aerosol particles with existing cloud droplets; Kipling et al., 2016; Flossmann et al., 1985) and is followed either by (1) precipitation that reaches the surface, removing the particle from the atmosphere (Radke et al., 1980), or (2) evaporation of cloud droplets or precipitation, returning the scavenged particle to the free atmosphere (Mitra et al., 1992). Model improvements in in-cloud scavenging include using a continuous rather than binary cloud fraction (Ryu and Min, 2022; Xu and Randall, 1996), accounting for cloud water phase (Grythe et al., 2017; Liu and Matsui, 2021), and accurately simulating cloud

supersaturation (Moteki et al., 2019). Although in-cloud scavenging is generally thought to be more efficient at removing accumulation mode aerosol particles (Watson-Parris et al., 2019; Choi et al., 2020), other studies argue that there are instances wherein below-cloud scavenging becomes more important at regulating aerosol burdens (Kim et al., 2021; Ryu and Min, 2022; Xu et al., 2019). Uncertainties related to wet scavenging are further exacerbated by the divergent role of clouds, which can be a sink or source of aerosol particles depending on environmental factors and cloud characteristics (Ryu et al., 2022).

One avenue for improving the estimation of wet scavenging, particularly in observational studies, is to identify an effective meteorological indicator of wet scavenging. Previous studies used precipitation amount (Feng, 2007; Andronache, 2003) while more recent studies accounted for air mass history using the National Oceanic and Atmospheric Administration (NOAA) Hybrid Single Particle Lagrangian Integrated Trajectory Model (Rolph et al., 2017; Stein et al., 2015) to calculate accumulated precipitation along trajectories (APT) (e.g., Kanaya et al., 2016, 2020). However, APT can be problematic as an indicator of wet scavenging because APT is an accumulated quantity and does not consider specific characteristics of precipitation relevant to scavenging such as intensity and frequency (Hou et al., 2018; Wang et al., 2021b, c; Hilario et al., 2022). APT as an indicator of wet scavenging also relies on the correct detection of precipitation and retrievals of amounts, which are challenging during both light (Nadeem et al., 2022; Kidd et al., 2021) and intense precipitation events (Chen et al., 2020a; Gupta et al., 2020) and even show disagreements between different satellite precipitation products (SPPs) and reanalyses (Cannon et al., 2017; Jiang et al., 2021; Alexander et al., 2020; Chen et al., 2020b; Barrett et al., 2020). Furthermore, precipitation from SPPs such as the Precipitation Estimation from Remotely Sensed Information using Artificial Neural Networks – Climate Data Record (PERSIANN-CDR) (Ashouri et al., 2015; Nguyen et al., 2018) and the Integrated Multi-satellitE Retrievals for the Global Precipitation Measurement (GPM) mission (IMERG) (Huffman et al., 2020) refer to total column precipitation that have been validated mainly with surface measurements (Sapiano and Arkin, 2009; Nicholson et al., 2019; Wang et al., 2021a) and consequently may not detect precipitation that evaporates before reaching the surface (e.g., virga) (Wang et al., 2018).

Given the uncertainties of estimating wet scavenging from precipitation, we present a flexible, computationally inexpensive method to identify alternative meteorological variables that can be used to better estimate wet scavenging. We combine curve-fitting and k-fold cross-validation to evaluate an array of meteorological variables from aircraft, satellite, and reanalysis data to answer the following:

(1) What meteorological variables can estimate wet scavenging trends better than APT? Since precipitation frequency has been shown to exert significant control over aerosol scavenging (Wang et al., 2021b), we hypothesize that predictors that account for the frequency of scavenging-conducive conditions (e.g., frequency of high relative humidity (RH) conditions along trajectories) will be able to capture wet scavenging trends better than APT.

(2) How can APT be filtered or changed to better estimate wet scavenging? We hypothesize that considering precipitation intensity and/or trajectory altitude thresholds when calculating APT will improve its ability to estimate wet scavenging. We also hypothesize that calculating APT using SPPs will perform better than APT from reanalysis.

The presented method may be repeated over different regions to identify region-specific wet scavenging indicators. This can inform scavenging parameterization development for models by providing guidance on what meteorological variables are needed to properly capture wet scavenging processes over a specific region. Future studies can also use the best-performing variables identified in this study as alternatives to APT when estimating the extent of wet scavenging.

## 2 Data & Methods

### 2.1 Aircraft data

Much of the methodology and instrument details in this study are detailed elsewhere (Hilario et al., 2021) but are summarized here. We utilize aircraft measurements from NASA's Cloud, Aerosol, and Monsoon Processes-Philippines Experiment (CAMP$^2$Ex; 24 August to 5 October 2019) over the tropical West Pacific (5 – 20°N, 117 – 127°E) (Reid et al., 2023), which hosts a dynamic transport environment rich in aerosol sources and cloud-precipitation systems.

Black carbon (BC)-equivalent concentrations (particle diameters: 100 – 700 nm; units: μg m$^{-3}$) were measured with a Single-Particle Soot Photometer (SP2) (Moteki & Kondo, 2007, 2010) with an uncertainty of 15% (Slowik et al., 2007) and lower detection limit of 10 ng m$^{-3}$ verified by filter-blank measurements as well as observations in the clean free troposphere. To eliminate in-cloud sampling artifacts such as droplet shattering on the inlet (Murphy et al., 2004), we use only data collected outside of clouds. All BC concentrations are reported at standard temperature and pressure (273 K, 1013 hPa). Carbon monoxide (CO; ppm) was measured using a dried-airstream near-infrared cavity ringdown absorption spectrometer (G2401-m; PICARRO, Inc.), with an uncertainty of 2% and precision of 0.005 ppm. As an in situ (i.e., at the aircraft's position) contrast to moisture-based variables along trajectories, relative humidity (RH$_{W, DLH}$) was derived from absolute water vapor concentrations that were retrieved by a diode laser hygrometer (DLH) (Livingston et al., 2008) on the aircraft.

### 2.2  Calculation of enhancement ratios

To relate wet scavenging to meteorological conditions during transport, previous studies calculated enhancement (Δ) ratios of BC and CO (ΔBC/ΔCO; Hilario et al., 2021; Kanaya et al., 2016; Oshima et al., 2012) which can then be used to quantify the transport efficiency of BC (TE$_{BC}$) (Kanaya et al., 2016, 2020; Oshima et al., 2012), discussed more in Sect. 2.5. By using the enhancement above a local background, ΔBC/ΔCO accounts for background concentrations of BC and CO at a receptor site ($\left(\frac{\Delta BC}{\Delta CO}\right)_{receptor}$) and is better able to detect a transported air mass containing BC and CO above local background levels. This ratio can be used as an indicator of wet scavenging because BC is relatively chemically inert and is mainly removed from the atmosphere via wet scavenging (Moteki et al., 2012). While CO is also relatively chemically inert, CO has a lifetime between 30 to 90 days (Seinfeld and Pandis, 2016) that is mainly controlled by photochemistry rather than wet scavenging due to its low solubility.

Enhancements were defined as the difference between species concentrations and the lowest 5[th] percentile species concentration for all CAMP$^2$Ex data for every 5 K potential temperature bin (Koike et al., 2003; Matsui et al., 2011). As CAMP$^2$Ex spanned the late southwest monsoon and early monsoon transition, background concentrations (i.e., lowest 5[th] percentile) were calculated for each monsoon phase using 20 September 2019 to divide the two monsoon phases. Only data with ΔCO > 0.02 ppm were included to reduce uncertainties caused by low denominator values in the ΔBC/ΔCO ratio (Kleinman et al., 2007; Kondo et al., 2011; Matsui et al., 2011). When calculating transport efficiency (Sect. 2.5), we converted ΔCO (from the receptor) from ppm to μg m$^{-3}$ using the ambient pressure and temperature measured by the aircraft such that ΔBC/ΔCO would be unitless.

As ΔBC/ΔCO is expected to vary by source region, Fig. S1a shows source-resolved distributions of ΔBC/ΔCO (unitless) based on source regions identified by Hilario et al. (2021), which classified backward trajectories into source regions using bounding boxes over major source regions established in previous literature. In addition to passing over source region bounding boxes, the source classification also considered (1) trajectory altitude, specifically whether or not the trajectory was below 2 km AGL which conservatively approximates climatological boundary layer heights over the region (Chien et al., 2019), as well as (2) trajectory residence time within each bounding box (minimum residence time: 6 hours). As described in Hilario et al. (2021), ΔBC/ΔCO is

higher for air masses coming from East Asia or the Maritime Continent (Fig. S1a), which suggests a low degree of aerosol scavenging during transport, while lower $\Delta BC/\Delta CO$ are seen for air from the Peninsular Southeast Asia, indicating scavenging had occurred. More information on major transport patterns affecting BC and CO during CAMP[2]Ex are provided in Appendix A.

### 2.3 Trajectory modeling

Trajectory modeling is a computationally inexpensive tool for characterizing transport processes (Kanaya et al., 2016; Oshima et al., 2012; Moteki et al., 2012) and has been used in synergy with aircraft data (Hilario et al., 2021; Dadashazar et al., 2021). In this study, we use trajectories to account for meteorological conditions during air parcel transport that are expected to impact the scavenged aerosol fraction. Backward trajectories were spawned every minute along the aircraft flight path and run for 72 hours using the NOAA HYSPLIT model. Meteorological input data for the HYSPLIT model were from the National Centers for Environmental Prediction (NCEP) Global Forecast System reanalysis (GFS; $0.25° \times 0.25°$). Figure S1b shows the distribution of transport times from different source regions (Sect. 2.2) to the CAMP[2]Ex aircraft. Generally, transit times are below 72 hours, indicated by 25[th] and 75[th] percentiles less than 72 hours, suggesting that 72 hours is sufficient to capture long-range transport from major source regions into the tropical West Pacific.

### 2.4. Emission inventory

To calculate the BC/CO emission ratio over each trajectory ($ER_{BC/CO}$), we used data from the Copernicus Atmosphere Monitoring Service (CAMS) Global Anthropogenic Emissions (CAMS-GLOB-ANT) inventory, version 5.3 (Soulie et al., 2023) which is based on the Emissions Database for Global Atmospheric Research (EDGAR) inventory from the European Joint Center (Crippa et al., 2018) and the Community Emissions Data System (CEDS) from the Joint Global Research Institute (Hoesly et al., 2018). CAMS-GLOB-ANT has a horizontal resolution of $0.1 \times 0.1°$ at monthly resolution. CAMS-GLOB-ANT accounts for 17 emission sectors, including shipping from CAMS-GLOB-SHIP v3.1, and emissions are reported in units of mass flux (kg m$^{-2}$ s$^{-1}$). More information on CAMS global and regional emissions can be found in Granier et al. (2019).

### 2.5. Calculation of transport efficiencies

The TE$_{BC}$ (unitless) was calculated for each trajectory using Eq. 1:

$$TE = \frac{\left(\frac{\Delta BC}{\Delta CO}\right)_{receptor}}{ER_{BC/CO}} \tag{1}$$

where $\left(\frac{\Delta BC}{\Delta CO}\right)_{receptor}$ is the enhancement ratio calculated from the aircraft data (Sect. 2.2) and $ER_{BC/CO}$ is the weighted-average emission ratio of BC/CO along each 72-h trajectory, inverse-weighted by altitude and calculated using emissions from the CAMS-GLOB-ANT inventory (Sect. 2.4). When calculating $ER_{BC/CO}$ for each trajectory, we applied a weighting function (Fig. S2a) to assign higher weights to lower altitudes such that the resulting $ER_{BC/CO}$ will be mainly determined by times when trajectory altitude is low, reflecting the higher likelihood of entraining surface emissions when the trajectory is close to the surface. An example of the weighting function as a function of trajectory altitude is shown in Fig. S2b wherein weighting decreases with increasing trajectory altitude. As the $ER_{BC/CO}$ calculation included the entire 72 hour length of the trajectory, our method of computing $ER_{BC/CO}$ is not restricted only to source regions (e.g., East Asia) but also accounts for potential entrainment of BC or CO over the open ocean, where sources such as shipping could contribute BC (Lack and Corbett, 2012) and CO (Jalkanen et al., 2012). We note that $ER_{BC/CO}$ is not required to be an enhancement ratio because the purpose of the enhancement ratio is to account for local background concentrations over the receptor region (Sect. 2.2).

We found that $TE_{BC}$ and $\Delta BC/\Delta CO$ are strongly correlated ($R^2 = 0.90$); however, $TE_{BC}$ has the added advantage of accounting for surface emissions of BC and CO that could have been entrained into transported air mass. $\Delta BC/\Delta CO$ is assumed to be influenced by two main factors: (1) source emissions of BC and CO along the trajectory path, and (2) removal of BC via wet scavenging. By setting $TE_{BC}$ as our predictand, we account for emissions encountered during long-range transport ($ER_{BC/CO}$) such that $TE_{BC}$ is expected to vary mainly via sinks (i.e., wet scavenging).

To show the variation of $ER_{BC/CO}$ and $TE_{BC}$ with different source regions, Fig. S1 shows source-resolved $ER_{BC/CO}$ (Fig. S1c) and $TE_{BC}$ (Fig. S1d). Air masses from East Asia show the smallest range in $ER_{BC/CO}$ while air masses from the Maritime Continent and Peninsular Southeast Asia have largely similar distributions. Lower values of $ER_{BC/CO}$ in air masses from the Maritime Continent are related to smoke from agricultural burning that coincided with the CAMP$^2$Ex period (Ge et al., 2014). Previous emission factor measurements showed that these fires tended to be smoldering rather than flaming, emitting CO but notably lower BC (Stockwell et al., 2015). While some variation is indeed present between source regions, the distributions of $ER_{BC/CO}$ are generally similar with modes between 0.22 – 0.26, which may explain the strong correlation between $TE_{BC}$ and $\Delta BC/\Delta CO$.

## 2.6 Data for predictor variables

Several meteorological variables (i.e., predictors) considered in this work were calculated from GFS reanalysis collocated along each trajectory. Though reanalysis is relatively coarse and not cloud-resolving, reanalysis variables (e.g., RH) may still be useful in detecting the presence of meso-to-synoptic-scale cloud fields. As precipitation is expected to be accompanied by elevated RH or water vapor mixing ratio (MR), these reanalysis-derived variables could serve as effective scavenging indicators in cases where precipitation may be missed or misestimated.

In addition to APT from GFS, we calculated APT from two SPPs: PERSIANN-CDR ($0.25° \times 0.25°$, daily resolution) (Ashouri et al., 2015; Nguyen et al., 2018) and IMERG Final v6 ($0.1° \times 0.1°$, 30-min resolution) (Huffman et al., 2020). We converted precipitation from these products to hourly amounts to match trajectory timesteps prior to further calculation.

Besides APT, we also calculated precipitation amount (PA; mm h$^{-1}$), frequency (PF), and intensity (PI; mm h$^{-1}$), which are well-established in the literature for characterizing precipitation, particularly in diurnal cycle analyses (e.g., Zhang et al., 2017; Hilario et al., 2020). Applying these quantities to precipitation along trajectories, PA is APT divided by the total number of hours along the trajectory (i.e., trajectory length) to obtain an average hourly precipitation rate, PF is the fraction of hours along the trajectory where the grid cell precipitation is above 0 mm h$^{-1}$, and PI is the ratio of PA to PF. Table 1 shows notation used to explain each type of predictor and its variations.

## 2.7 Curve-fitting and k-fold cross-validation

To quantify relationships between $TE_{BC}$ and each predictor as well as its uncertainty, we performed k-fold cross-validation (k = 10) parallelized using the Python package Jug, version 2.2.2 (Coelho, 2017). To create k distinct partitions of the data, we utilized stratified random sampling wherein random sampling was performed for each 5$^{th}$ percentile block of the predictor such that the sampling probability better reflects the distribution of predictor values, which is important for skewed distributions such as precipitation amount, and the resulting k partitions capture the behavior of $TE_{BC}$ across the full spectrum of predictor values. By randomly sampling each percentile block for k distinct partitions, this sampling method improves the chances of capturing intra-block variability in $TE_{BC}$ by collecting the most samples where the highest data coverage exists. The random nature of the sampling also allows for the consideration of extreme values in the curve-fitting, with a sampling probability proportional to the frequency of these extreme values. As an example, Fig. 1a-b shows the emphasis of the stratified random sampling method on high density areas of the scatterplot of $RH_{q90}$ and $TE_{BC}$, denoted by dense percentile blocks (gray dashed lines). For extremely skewed

distributions such as APT, several of the lower-value percentiles exhibited non-unique values (e.g., zero). In the case of repeated percentile values, these percentile-based groups were merged. We imposed a minimum of six distinct percentile blocks to ensure robust curve-fitting.

An iterative train-test split procedure using these partitions was then performed using k-1 partitions as the training set and the remaining partition as the testing set (Fig. 1a). For each iteration of the k-fold cross-validation, nonlinear least squares curve-fitting was applied to the training set (i.e., 9 partitions for a total k of 10) to determine coefficients for the equation (e.g., general exponential; discussed below) fitted onto the scatterplot of $TE_{BC}$ and the predictor. We used these coefficients and the testing set (i.e., the remaining partition) as inputs for the curve-fitting equation and calculated a predicted curve of $TE_{BC}$ and the predictor. To assess this predicted curve, we applied stratified random sampling on the testing set and took the median $TE_{BC}$ per $5^{th}$ percentile block to create observed curves of $TE_{BC}$ as a function of the predictor that could be compared to the predicted curve (Fig. 1b). Because decreases in $TE_{BC}$ are expected to be mainly from wet scavenging, the overall trend or median curve may be treated as a reasonable indicator of wet scavenging effects on $TE_{BC}$ related to changes in the predictor value.

Using a linear regression of predicted and observed median $TE_{BC}$ per 5th percentile block of the predictor (Fig. 1c), we calculated statistics (e.g., slope, R) to describe how well the predictor can predict $TE_{BC}$. Specifically, the performance of a predictor refers to how well $TE_{BC}$ derived from the predictor matches observed $TE_{BC}$. We also computed statistics comparing predicted and observed $TE_{BC}$ for individual points (Fig. 1d) rather than medians to assess how much variability in $TE_{BC}$ is captured by the predicted curves. The population in Fig. 1d visually follows the 1-to-1 line, indicating good performance of the model; however, the best-fit line on individual points was greatly affected by outlier points of high observed $TE_{BC}$ that led to poor agreement between the best-fit and 1-to-1 lines when the actual agreement was much better (visually). This suggests that the median-based statistics (Fig. 1c) are more robust to outliers and present a fairer evaluation of model predictions. Note that the individual-point statistics (Fig. 1d) resulted in correlations and slopes further from ideal values compared to the median-based statistics. This is expected as individual $TE_{BC}$ data points exhibit large variability due to the influence of factors other than wet scavenging; however, a comparison of our results when using individual-point or the median-based statistics show that they agree quite well qualitatively, with the relative ranking of predictors largely unchanged between the two types of statistics. In other words, the top predictors performed well whether we used median-based or individual-point statistics, implying the conclusions reached using our method are qualitatively insensitive to this choice. For simplicity, reported statistics in this study refer to median-based statistics unless otherwise specified.

To determine if a predictor tended to overestimate or underestimate $TE_{BC}$, we calculated a weighted area difference (WAD) using Eq. 1:

$$WAD = \frac{\sum N_i \cdot x_i}{\sum N_i} \tag{1}$$

where $x_i$ is the difference between observed and predicted $TE_{BC}$ for the $i^{th}$ percentile block and $N_i$ is the number of data points in that percentile block. A positive (negative) WAD indicates an overestimate (underestimate) of observed $TE_{BC}$.

To account for differing relationships between $TE_{BC}$ and each predictor, we applied curve-fitting on the scatterplot of each predictor and $TE_{BC}$ using multiple nonlinear equations (Table 2) and chose the equation that produced the highest Pearson correlation (R) between observed and predicted $TE_{BC}$ for that predictor. We considered equations from two previous studies (Kanaya et al., 2016; Oshima et al., 2012) and two generalized equations (Gaussian, general exponential) to capture other types of relationships (Table 2). The inclusion of the latter two are to account for a wider range of possible relationships between predictors and $TE_{BC}$, such as the case of a predictor capturing $TE_{BC}$ trends well but not having an inversely proportional relationship with $\Delta BC/\Delta CO$. The case of a non-inversely proportional relationship with $TE_{BC}$ is still interesting because a strong relationship implies

that the variable could be used to estimate $TE_{BC}$ even if their relationship is nonmonotonic. The final assigned equations led to similar root mean squared error (RMSE) across predictors (Fig. S3) suggesting it is fair to compare different predictors.

      When selecting which equation to use (among those in Table 2) for fitting between the predictor and $TE_{BC}$, we opted for the equation that resulted in the highest R between observed and predicted $TE_{BC}$ (e.g., Fig. 1c). The basis of this choice on R was because our objective is to identify predictors that can at least capture trends in $TE_{BC}$. After selecting which equation to use per
predictor, the subsequent comparison (Sect. 3) of the performance of different predictors considers other statistical metrics such as slope, intercept, and WAD. For some combinations of predictors and equations, the curve-fitting did not successfully converge (< 4% of all combinations and k-fold iterations). In these cases, we did not include the predictor-equation combination in our analysis. However, curve-fitting on the predictor may still converge when using a different equation. In such a scenario, the predictor becomes part of our analysis.

Sensitivity testing with the k value showed no significant effect on the general conclusions of the study when k was changed between 5 and 20 (not shown). We opted for k = 10 based on previous work evaluating different accuracy estimation methods which showed that k = 10 is sufficient to estimate performance metrics (e.g., $R^2$) while minimizing computational expense (Breiman and Spector, 1992; Kohavi, 1995).

      Although there is no physical process built into this procedure, the strength of the method is its repeatability in different
environments or regions with minimal changes to the overall procedure. As it requires no physical model to be run besides the trajectory calculations, the method is also relatively computationally inexpensive. Future work wanting a more physical basis may apply our method as a diagnostic tool to identify and narrow down a list of meteorological variables that may be relevant to wet scavenging and continue their analysis with a physical model using the narrowed list of variables to analyze.

## 3 Results and Discussion

**3.1 Overall statistical performance**

      Figures 2, 3, and S3 show performance comparisons of different predictors derived from linear regressions of observed and predicted $TE_{BC}$. Hereafter, the performance of a predictor in this study refers to a predictor's ability to predict observed $TE_{BC}$ based on curve-fitting (Sect. 2.7; Fig. 1c). To simplify these figures, only the top eight predictors per panel (by R) are colored to focus our discussion on predictors that were able to predict $TE_{BC}$. Table S1 provides the equation and coefficients used for the top eight
predictors per panel of Fig. 2.

      Using APT-based predictors (Fig. 2a) led to moderate R between predicted and observed $TE_{BC}$ but slopes far below the ideal value of 1, which, in addition to positive intercepts and WAD (Fig. S3a), indicate that APT-based predictors tend to underestimate $TE_{BC}$ when APT is high.

      In comparison, predictors in Fig. 2b are based on RH (e.g., $f_{RH95}$, $RH_{q90}$) or MR (i.e., $f_{MR15}$) and predicted $TE_{BC}$ much better
in terms of trends (high R) and magnitude (slopes close to 1), suggesting that these predictors (Fig. 2b) could be better at estimating $TE_{BC}$ than APT (Fig. 2a). One possibility for this is that APT is an accumulated value that does not account for frequency or intensity, both of which have been argued to be important for regulating aerosol scavenging (Hou et al., 2018; Wang et al., 2021c). To explore this possibility further, we calculated PA, PF, and PI for each trajectory (Figs. 2c-e). Among these three, PF (Fig. 2d) and PI (Fig. 2e) resulted in the best slopes and R, with PI showing slightly better slopes and R than PF. In comparison, PA
performed poorly, similar to APT, which is expected as both are related to summed precipitation amount. Comparing the PI variables with the highest R (i.e., colored points in Figs. 2e), the majority of these good-performing PI variables were filtered for heavier or more intense precipitation (i.e., > 0.2 mm). This filtering for heavier precipitation was done by including only grid cells

with precipitation > 0.2 mm when calculating PI. A similarly good performance was observed for PF variables that also filtered for more intense precipitation. These results suggest that PI (and to a lesser degree PF) must be accounted for when predicting aerosol scavenging over the tropical West Pacific. This further implies that even though precipitation may be occurring, it may not be efficiently scavenging aerosol.

Comparing which precipitation products among the top predictors (by R), most good-performing precipitation-based predictors used SPP-based precipitation such as IMERG or PERSIANN-CDR (Fig. 2c-e). This suggests that GFS-derived precipitation variables are not as able to capture observed $TE_{BC}$ trends. The poor performance of GFS-derived precipitation is reflective of past studies showing disagreements in precipitation characteristics between satellite and reanalyses (Cannon et al., 2017; Jiang et al., 2021) and even divergent precipitation trends and amounts among individual reanalysis products (Alexander et al., 2020; Chen et al., 2020b; Barrett et al., 2020). Our results corroborate previous work that precipitation from GFS reanalysis is not a reliable predictor of aerosol scavenging compared to precipitation from SPPs. Future studies relating precipitation to aerosol scavenging are recommended to instead rely on in situ or satellite retrieved precipitation rather than precipitation from reanalysis.

Predictors based on quantiles of RH (e.g., $RH_{q90}$) (Fig. 2b) perform quite well, with high R and slope (Fig. 2e) and intercept and WAD consistently close to 0 regardless of quantile (Fig. 3b). $RH_{q90}$ performs slightly better in terms of R than other RH thresholds (Fig. 2b); however, this difference is minor as shown by the overlapping $25^{th}$-$75^{th}$ percentile error bars between the different RH quantiles. The similar performance between different RH quantiles suggests consistency in their ability to predict $TE_{BC}$ trends (high R; Fig. 2b) while doing reasonably better than other types of predictors when estimating $TE_{BC}$ magnitudes across the spectrum of predictor values (intercepts closer to 0, slopes closer to 1). Maximum RH along trajectories was used by Kanaya et al. (2016) in their analysis of $\Delta BC/\Delta CO$ scavenging to detect the role of clouds in BC removal. Our findings suggest that top quantiles of RH, including its maximum, are good choices for estimating $TE_{BC}$.

Compared to variables directly linked to precipitation (PA, PF, PI, APT), the slopes from RH quantiles are noticeably closer to the ideal value of 1 (Fig. 2b) while their intercepts are closer to the ideal value of 0 (Fig. 2b), meaning $TE_{BC}$ predicted by RH quantiles more closely matches the observed $TE_{BC}$ than $TE_{BC}$ predicted by precipitation. We hypothesize that the better performance of RH-related predictors over those more directly related to precipitation (e.g., APT) may be explained by instances of precipitation that are missed (or misestimated) by SPP retrievals that are indirectly detected by reanalysis as high humidity conditions. This possibility is supported by previous literature showing the tendency of SPPs to misestimate light (Nadeem et al., 2022; Kidd et al., 2021) or intense precipitation (Chen et al., 2020a; Gupta et al., 2020); however, we cannot rule out the possibility of the hygroscopic growth, in-cloud activation (high RH), and subsequent removal of BC during transport. Thus, our hypothesis of RH from reanalysis capturing missed precipitation from SPPs requires further investigation in future work.

Of all the fractional predictors considered in this study, $f_{MR15}$ and $f_{RH95}$ perform the best (Fig. 2b). $f_{MR15}$ and $f_{RH95}$ reflect the frequency of occurrence of scavenging-conducive conditions during transport. A high frequency of high MR or RH may indicate that air masses passed through large areas of clouds and/or precipitation during long-range transport. $f_{RH95}$ has a median slope of 0.99 (Fig. 2b), a $25^{th}$-$75^{th}$ percentile range in slope of 0.92 – 1.02 (Fig. 2b), and a median intercept of 0.01 (Fig. 3b), indicating that $f_{RH95}$ can be used to capture $TE_{BC}$ trends and magnitudes for a wide range of $TE_{BC}$. The good performance of $f_{MR15}$ and $f_{RH95}$ suggests that the frequency of scavenging-conducive conditions may be more reliable indicators of aerosol scavenging than precipitation amount (e.g., APT, PA).

### 3.2 Nonlinear sensitivity of $TE_{BC}$ to meteorological variables

Although the predictors in Figs. 2 – 3 exhibit the highest R of all predictors considered in this study, their slopes are generally below 1 (Fig. 2) while their intercepts and WAD are generally positive (Fig. 3). The combination of these statistics implies that

predictions of $TE_{BC}$ using our method tend to overestimate observed $TE_{BC}$ across the spectrum of predictor values (indicated by WAD > 0) with maximum overestimations occurring when observed $TE_{BC}$ is low (indicated by slopes < 1 and intercepts > 0). This points to a nonlinear sensitivity of $TE_{BC}$ to these predictors as the degree of scavenging increases. Dadashazar et al. (2021) observed a similar nonlinear response to APT by a ratio of particulate matter below 2.5 µm to CO ($\Delta PM_{2.5}/\Delta CO$), where $\Delta PM_{2.5}/\Delta CO$ was most responsive to APT when APT was below 5 mm and less sensitive to APT when APT exceeded 5 mm.

Investigating this sensitivity further, Fig. 4 shows that PF-predicted $TE_{BC}$ does not capture the trends of observed $TE_{BC}$ for highly scavenged air masses. In other words, PF loses its predictive power as the degree of scavenging increases, implying that PF is most important for the scavenging of fresher air masses (high-$TE_{BC}$). This nonlinear sensitivity of $TE_{BC}$ to PF hints at the possibility that other meteorological variables may become important for further scavenging of highly scavenged air (low-$TE_{BC}$). In contrast to predictors directly related to precipitation (Fig. 4d-f), the predicted curves of $RH_{q95}$ (Fig. 4a), $f_{RH95}$ (Fig. 4b), and $f_{MR15}$ (Fig. 4c) visibly track the trends of observed $TE_{BC}$ with approximately half the difference between predicted and observed $TE_{BC}$ when $TE_{BC}$ is low. The capability of $RH_{q95}$, $f_{RH95}$, and $f_{MR15}$ to predict $TE_{BC}$ across a wider range of values is further reflected by generally lower intercepts and WAD (Fig. 3) than precipitation-related predictors, which suggests promising alternative indicators of aerosol scavenging. However, we also note that such differences could arise partly from the limitations of curve-fitting, wherein fitted curves naturally capture gradual changes (e.g., Fig. 4b) better than sharp ones (e.g., Fig. 4d).

### 3.3 Applying filters to improve the predictive power of precipitation-related variables

In this section, we examine the predictive power of precipitation-related variables when applying the following filters: (1) precipitation intensity, (2) trajectory altitude, (3) data product, and (4) trajectory length, with the objective to identify what factors are important when relating precipitation along trajectories to $TE_{BC}$. Filtering for precipitation intensity isolates the contribution of higher precipitation intensities towards a precipitation-related predictor's ability to predict $TE_{BC}$. Intense precipitation has been shown to be more efficient at scavenging aerosol particles (Zhao et al., 2020) and may be important when estimating aerosol scavenging. Filtering for trajectory altitude (i.e., considering precipitation only when the trajectory altitude is below 1.5 km AGL) tests the hypothesis that air masses within the boundary layer will be most susceptible to wet scavenging. Grythe et al. (2017) demonstrated that below-cloud scavenging (i.e., impaction by precipitation) accounted for majority of scavenging events below 1 km. We selected 1.5 km based on previous work on the marine boundary layer over the tropical West Pacific (Chien et al., 2019). We repeated the analysis for three precipitation products (one reanalysis and two SPPs) to capture variability in our results due to the choice of data product which has been shown to be important for precipitation (Alexander et al., 2020). Finally, we tested the effect of trajectory length on the performance of APT as a predictor of $TE_{BC}$. We performed these sensitivity tests on APT (Fig. 5), PI (Fig. S4), PF (Fig. S5), and PA (Fig. S6).

In general, we found that applying altitude and/or precipitation filters negatively affected the performance of APT (Fig. 5b-d), PF (Fig. S5b-d, except for PERSIANN-CDR), and PA (Fig. S6b-d), leading to lower R between predicted and observed $TE_{BC}$ compared to the case without any filters (Fig. 5a). Two exceptions were: PF from PERSIANN-CDR (colored yellow in Fig. S5), which could be used to estimate $TE_{BC}$ if we applied both an altitude filter (< 1500 m) and a precipitation intensity filter (> 0.2 mm h$^{-1}$) over longer back trajectory times (> 48 hours) (Fig. S5d), and PI, which could be used to estimate $TE_{BC}$ when filtering for precipitation intensities (> 0.2 mm h$^{-1}$) and along trajectories longer than 48 hours (Fig. S4b). The better performance of PI across multiple SPPs (Fig. S4) is an encouraging sign that this improvement is robust and points to the contribution of higher precipitation intensities towards total scavenging.

Applying a filter for trajectory altitude prior to calculating APT also did not lead to large improvements in R (Fig. 5c). This was surprising because, when using total column precipitation from SPPs, a maximum altitude filter should reduce errors from

cases where precipitation occurs below the air mass and no scavenging occurs. Since the SPPs used in this study have been validated using surface measurements (Sapiano and Arkin, 2009; Nicholson et al., 2019; Wang et al., 2021a), precipitation from SPPs should be reflective of precipitation that reaches the surface, implying a susceptibility of these SPPs to errors related to virga (Wang et al., 2018). However, Wang et al. (2018) also showed that virga occurrence over the tropical West Pacific is also infrequent. An alternative explanation for the poor performance of altitude-filtered APT is uncertainties related to trajectory altitude (Harris et al., 2005), such that an air parcel may have actually been traveling at a lower altitude than its modelled trajectory and underwent more scavenging than predicted using APT. An examination of trajectory altitudes (Fig. S7) revealed that filtering for trajectory altitudes below 1.5 km excluded the majority (~70%) of precipitating grid cells encountered by trajectories, which likely negatively impacted the predictive ability of altitude-filtered predictors.

Longer trajectories resulted in slightly higher R between observed and predicted $TE_{BC}$ using APT from PERSIANN-CDR or IMERG (Fig. 5a-b); however, this difference is not large, as shown by overlapping $25^{th}$-$75^{th}$ percentile error bars. Interestingly, this increase in R for longer trajectories was more evident when filtering for precipitation intensities $> 0.2$ mm h$^{-1}$ when calculating APT (Fig. 5b) or PI (Fig. S4b), but not when applying this intensity filter on PF (Fig. S3b) or PA (Fig. S4b). Further interpretation likely requires a physical model in future work to explain why the performance of intense precipitation (Fig. 5b) benefits from a longer trajectory more than total precipitation does (Fig. 5a).

## 4 Limitations

**$TE_{BC}$ as a wet scavenging proxy**: In this study, we treat $TE_{BC}$ as a proxy for wet scavenging (i.e., predictand) and base our conclusions on which variables (i.e., predictors) best predict $TE_{BC}$. Dilution or entrainment during transport is expected to influence the $\Delta BC/\Delta CO$ ratio and therefore $TE_{BC}$. While the use of the CAMS-GLOB-ANT emission inventory (Sect. 2.4) when calculating $ER_{BC/CO}$ (Sect. 2.5) reduces this uncertainty by accounting for potential surface influence during transport close to the surface, the resolutions of both the trajectory meteorological input ($0.25 \times 0.25°$) and the emission inventory ($0.1 \times 0.1°$) remain limiting factors. Thus, our analysis assumes that wet scavenging is the main driver of changes in $TE_{BC}$ and chemical transport modelling in future work is needed to quantify the effect of mixing on $TE_{BC}$. Consequently, this method is expected to work well in outflow regions such as the tropical West Pacific and not well where additional BC and/or CO are likely to be after initial emission or wet scavenging has occurred (e.g., continental region).

**Method does not discriminate between in- or below-cloud scavenging:** The conclusions of our study are based on the relative ability of different variables to predict $TE_{BC}$, our proxy for wet scavenging. This approach does not isolate individual processes that are usually parameterized by global circulation models (e.g., impaction, nucleation) (Croft et al., 2009, 2010; Ryu and Min, 2022) and does not discriminate between in-cloud or below-cloud scavenging. However, through our proposed framework, we can still gain qualitative insights into which meteorological variables are relevant for estimating aerosol scavenging, which can inform future studies as well as developments in model parametrization.

**Single predictor method:** The method presented here assesses the one-to-one relationship between a single predictor and $TE_{BC}$ repeated individually for several predictors. We expect that using a combination of predictors may lead to better predictions of $TE_{BC}$ while providing a more physical picture of relative contributions of different meteorological variables towards wet scavenging. Future work may utilize multiple linear regression or more sophisticated methods such as machine learning to consider different combinations of predictors with the objective of identifying a combination that predicts $TE_{BC}$ well and extracting further information on what physical mechanisms may be relevant for the removal of $TE_{BC}$ based on relative coefficients or weightings of different predictors.

**Curve-fitting**: The results can depend on the curve-fitting function used. Different variables are expected to have different relationships with $TE_{BC}$. Thus, considering only one function for curve-fitting favors variables that have a specific relationship with $TE_{BC}$. To reduce this bias, we applied four different curve-fitting functions (Table 2) on each predictor based on two equations from previous studies (Kanaya et al., 2016; Oshima et al., 2012) and two equations of generalized form that accounted for possible relationships between $TE_{BC}$ and each predictor. We then chose the curve-fitting function that produced the highest R between observed and predicted $TE_{BC}$. However, we note that this does not completely remove the bias as specific functions were still selected.

**Trajectory modeling:** Vertical motion through convection, entrainment, and detrainment processes are known uncertainties in trajectory modeling, which increase with trajectory length (Harris et al., 2005). The spatial and temporal resolutions of the meteorological input used for the HYSPLIT model are also limiting factors as meteorology along HYSPLIT trajectories do not account for sub-timestep or sub-grid processes.

## 5 Conclusions

We present a method to identify meteorological indicators of aerosol scavenging using a combination of aircraft, satellite, and reanalysis data coupled with HYSPLIT backward trajectories. We apply this method to the CAMP$^2$Ex field campaign over the tropical West Pacific, which hosts a wide range of cloud fractions and precipitation characteristics as well as an environment characterized by long-range transport of aerosol and trace gas species. We evaluate which meteorological variables can be used to predict $TE_{BC}$ (i.e., predictors). The main conclusions of the study are the following:

1. Although APT has been utilized in several studies as an indicator of aerosol scavenging, we demonstrate that APT does poorly when predicting $TE_{BC}$ (e.g., weak correlations, underestimates $TE_{BC}$). Furthermore, the application of altitude or precipitation intensity filters negatively impact the performance of APT in predicting $TE_{BC}$. Since APT is an accumulated precipitation amount over the whole trajectory, APT does not account for other precipitation characteristics such as intensity or frequency, which have been shown to be relevant for aerosol scavenging. This shortcoming may explain the overall poor relative performance of APT in predicting $TE_{BC}$.

2. Predictors based on specific quantiles of RH (e.g., $RH_{q90}$) also perform quite well in predicting both $TE_{BC}$ trends and magnitudes (intercepts close to zero, WAD close to zero, slopes close to 1, R close to 1). We find only minor differences in the performance depending on the exact quantile used, suggesting the RH distribution during transport is a robust way to estimate $TE_{BC}$. We hypothesize the outperformance of RH quantiles over predictors more directly related to precipitation (e.g., APT) to be due to missed precipitation in SPP retrievals that was indirectly represented in reanalysis as high humidity conditions; however, further work is required to explore this possibility.

3. Frequency-related predictors such as $f_{MR15}$ and $f_{RH95}$ perform better than APT in predicting $TE_{BC}$ trends (higher R) and magnitudes (slopes closer to 1). $f_{MR15}$ and $f_{RH95}$ represent the frequencies along 72-h trajectories of MR exceeding 15 g kg$^{-1}$ and RH exceeding 95%, respectively. The abilities of $f_{MR15}$ and $f_{RH95}$ to predict $TE_{BC}$ suggests that the frequency of humid conditions should be considered when estimating aerosol scavenging.

4. To investigate which precipitation characteristics are most relevant for predicting $TE_{BC}$, we quantify PA, PF, and PI along trajectories and find that PI is the most effective at estimating $TE_{BC}$ when we calculate PI over longer trajectories ($> 48$ h) and only include grid cells with precipitation $> 0.2$ mm h$^{-1}$ in our calculation. This points to the contribution of intense precipitation and the importance of accounting for air mass history when estimating aerosol scavenging.

5. We find that precipitation from SPPs (IMERG, PERSIANN-CDR) is generally better at predicting $TE_{BC}$ (higher R) than precipitation from reanalysis (GFS). This is corroborated by previous studies that found larger misestimates of precipitation by reanalysis than by SPPs. Because of our results and those of past studies, we recommend relying on in situ or SPP precipitation rather than precipitation from reanalysis, particularly when relating precipitation to aerosol scavenging.

Given these findings, we recommend the following alternatives to APT when estimating aerosol scavenging: (1) RH quantiles (e.g., 90[th] percentile of RH along trajectories), (2) $f_{MR15}$ or $f_{RH95}$, and (3) PI from SPPs filtered for grid cells with precipitation > 0.2 mm h$^{-1}$. These variables were found to be able to predict $TE_{BC}$ more accurately than APT; thus, future scavenging parametrizations should consider these meteorological variables along air mass histories.

Future work is encouraged to apply this method over a variety of environments (e.g., using other data from other field campaigns), utilize machine learning to assess what combinations of meteorological variables are relevant for predicting aerosol scavenging, and apply this method to other regions to determine if there are regional differences in indicators of aerosol scavenging. Furthermore, CAMP$^2$Ex included a rich dataset on cloud water composition (Crosbie et al., 2022; Stahl et al., 2021) that can be explored, as in past work for other regions (MacDonald et al., 2018), to gain additional insights into aerosol wet scavenging processes.

**Appendix A: Describing the transport of BC and CO during CAMP$^2$Ex**

During the CAMP$^2$Ex field campaign, BC and CO originated from several sources. Long-range transport patterns during the campaign and associated air mass composition are described in Hilario et al. (2021) but here we present a summary of their findings related to the transport of BC and CO. The CAMP$^2$Ex field campaign overlapped with the end of the southwest monsoon and the beginning of the monsoon transition (Reid et al., 2023). Because of this, a synoptic shift occurred during the campaign (Hilario et al., 2021; their Figs. 2-3) that allowed for the sampling of transported air masses from different source regions such as East Asia and the Maritime Continent. Hilario et al. (2021) identified four major source regions for long-range transport: East Asia (e.g., China, Korea), the Maritime Continent (e.g., Indonesia, Malaysia), Peninsular Southeast Asia (e.g., Vietnam), and the West Pacific (i.e., ocean). The presence of long-range transport was detected throughout the whole campaign (their Fig. S2).

The Maritime Continent during the campaign was undergoing its burning season which is well-established in the literature to lead to high aerosol loadings that can be transported over large distances (Xian et al., 2013). Hilario et al. (2021) showed that air masses from the Maritime Continent and East Asia were transported under relatively dry conditions, which in this study manifested as higher $\Delta BC/\Delta CO$ (Fig. S1a) and $TE_{BC}$ (Fig. S1d), and were associated with southwesterly monsoon flow and the passage of typhoons, respectively. These conditions were conducive for long-range transport and led to the sampling of higher concentrations of BC and CO in air from East Asia (BC: 87.29 ng m$^{-3}$; CO: 0.16 ppm) and the Maritime Continent (BC: 71.81 ng m$^{-3}$; CO: 0.18 ppm) than in air from Peninsular Southeast Asia (BC: 24.90 ng m$^{-3}$; CO: 0.10 ppm) or the West Pacific (BC: 1.03 ng m$^{-3}$; CO: 0.08 ppm). We note that Hilario et al. (2021) kept CO in units of ppm while we converted CO to mass concentration units such that $\Delta BC/\Delta CO$ would be unitless. Hilario et al. (2021) demonstrated that the scavenging of air from Peninsular Southeast Asia was related to convective lofting as air from Peninsular Southeast Asia sampled in the free troposphere (> 1.5 km) had much lower aerosol concentrations than air from the region sampled in the boundary layer (< 1.5 km) (their Fig. S6). These findings point to the active role of scavenging in determining aerosol loadings in transported air masses during CAMP$^2$Ex.

*Code availability.* Codes are freely available upon request to the authors.

*Data availability.* All datasets are publicly available and accessible at

https://doi.org/10.5067/Suborbital/CAMP2EX2018/DATA001 (NASA/LaRC/SD/ASDC, 2020). HYSPLIT data are accessible
through the NOAA READY website (https://www.ready.noaa.gov/index.php, last access: 11 April 2023) (NOAA Physical
Sciences Laboratory, 2020).

*Author contributions.* EC, LDZ, MAS, JPD, GSD contributed to data collection. MRAH and AS conceptualized the study. MRAH
performed the data analysis and prepared the manuscript with input from all co-authors.

*Competing interests.* The authors declare that they have no conflict of interest.

*Disclaimer.* Publisher's note: Copernicus Publications remains neutral with regard to jurisdictional claims in published maps and
institutional affiliations.

*Special issue statement.* This article is part of the special issue "Cloud Aerosol and Monsoon Processes Philippines Experiment
(CAMP²Ex) (ACP/AMT inter-journal SI)". It is not associated with a conference.

*Acknowledgements.* This research has been supported by the National Aeronautics and Space Administration (grant no.
80NSSC18K0148).

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

**Table 1: Examples of notation used in this study.**

| Predictor | Description | Variations |
|---|---|---|
| $f_{RH95}$ | Fraction of hours along trajectories where GFS relative humidity (RH) > 95% | $f_{RH85}$, $f_{RH90}$ |
| $f_{MR15}$ | Fraction of hours along trajectories where GFS water vapor mixing ratio (MR) > 15 g kg$^{-1}$ dry air | $f_{MR17}$ |
| $RH_{q95}$ | 95$^{th}$ percentile of RH along trajectories | $RH_{q50}$, $RH_{q85}$, $RH_{q90}$, $RH_{q100}$, $RH_{mean}$ |
| $RH_{w, DLH}$ | RH over water measured by DLH onboard the aircraft | - |
| $APT_{PCP > 0.2\ mm,\ 48H,\ GFS,\ < 1500m}$ | Accumulated precipitation calculated along 48-h trajectories where GFS precipitation is above 0.2 mm and trajectory altitude is below 1500 m | Trajectory duration: 12H, 24H, 48H, 72H<br>Precipitation product: GFS, IMERG, PERSIANN-CDR<br>Maximum altitude filter: no filter, < 1500 m<br>Minimum precipitation filter: no filter, > 0.2 mm<br>Other precipitation variables: PA, PF, PI |

**Table 2: Curve-fitting equations considered where x is the predictor variable and y is the observed ΔBC/ΔCO while a, b, c, and d are best-fit parameters determined via least-squares regression.**

| Name | Equation | Source |
|------|----------|--------|
| Gaussian | $y = a \cdot \exp(-\frac{(x-b)^2}{2 \cdot c^2}) + d$ | - |
| General Exponential | $y = a \cdot exp(-b \cdot x) + c$ | - |
| Oshima | $y = b - a \cdot \log_{10}(x)$ | Oshima et al. (2012) |
| Kanaya | $y = c \cdot exp(-a \cdot x^b)$ | Kanaya et al. (2016) |

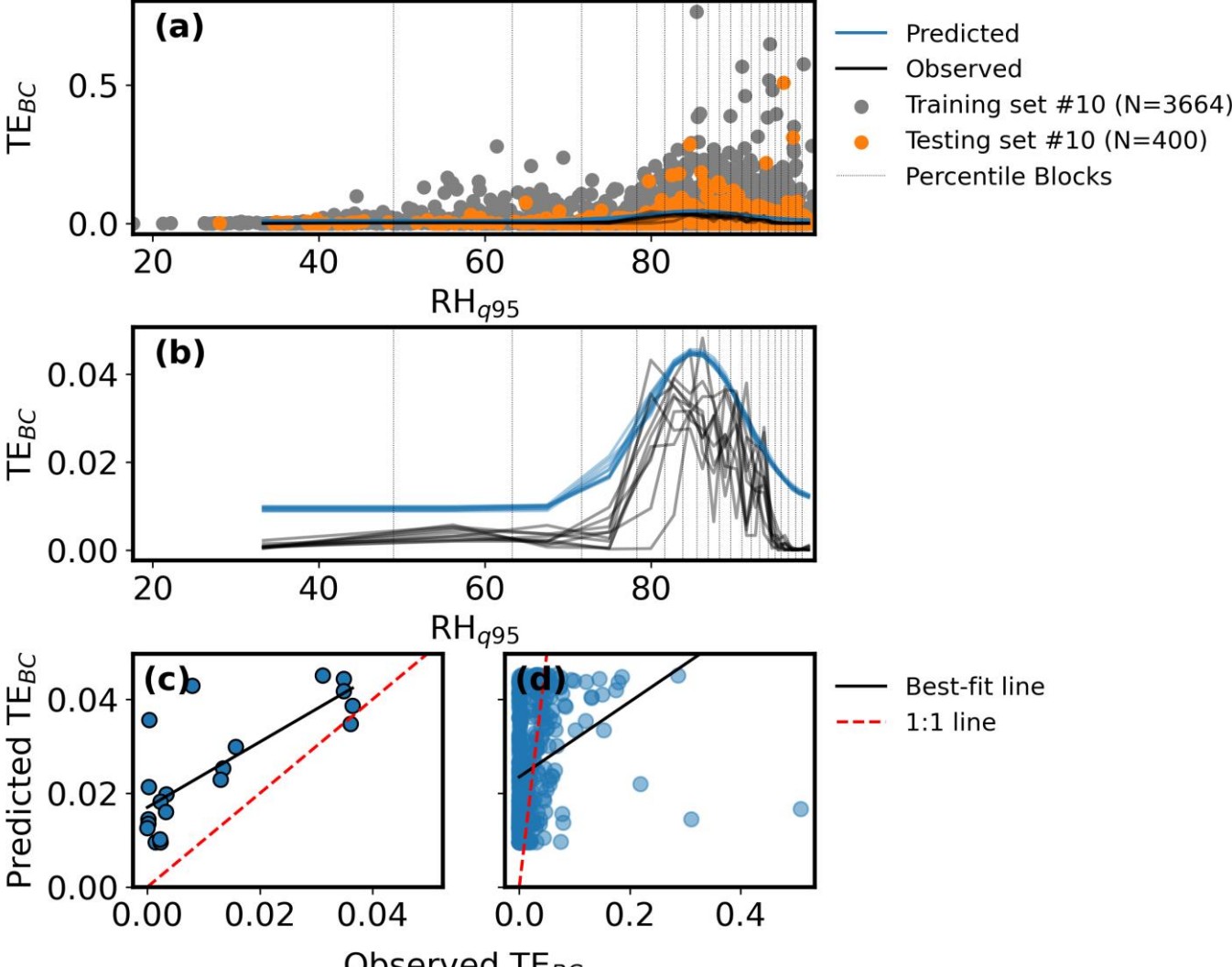

**Figure 1: An example of the curve-fitting procedure on TE$_{BC}$ with RH$_{q95}$ as the predictor fitted with a Gaussian function. (a) Training (gray dots) and testing sets (orange dots) for the 10$^{th}$ iteration of the k-fold cross-validation procedure selected using stratified random sampling. Percentile blocks of each X-axis variable are denoted by vertical gray lines with observed (black) and predicted curves (blue) also plotted for all 10 iterations. (b) Same as (a) but only showing observed (black) and predicted curves (blue) for all 10 iterations to highlight variations between the k iterations. (c) Scatterplot comparing RH$_{q95}$-predicted ΔBC/ΔCO and observed median TE$_{BC}$ per 5th percentile block of the predictor. Note that (c) is simply the linear regression of the observed and predicted curves in (b). (d) Same as (c) but comparing RH$_{q95}$-predicted and observed TE$_{BC}$ for individual points. In (c-d), only training set data are used, Y-axes are the same, the best-fit line is shown as a black line, and the 1:1 line is the red, dashed line.**

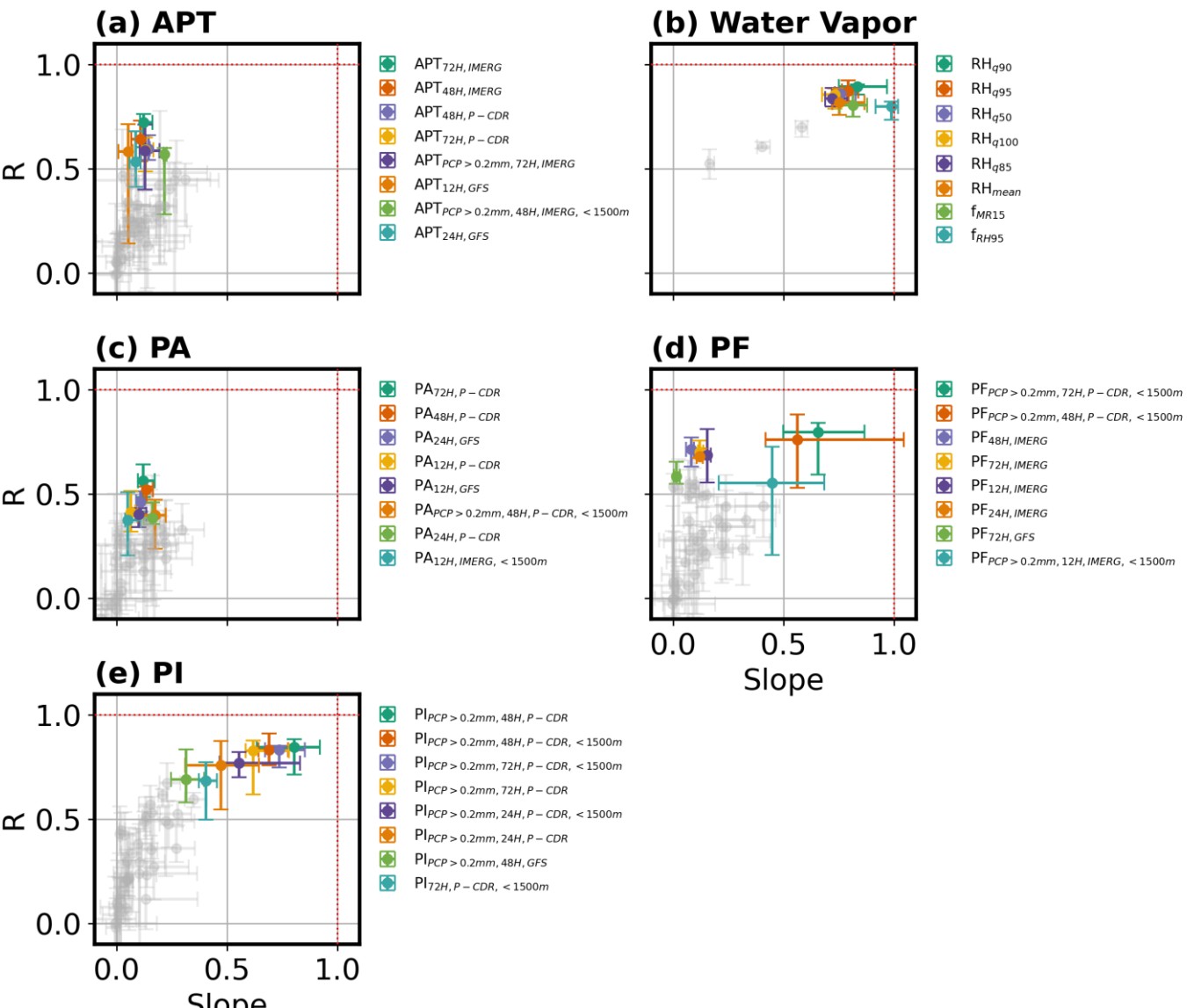

**Figure 2: Slope and Pearson correlation (R) values derived from linear regressions of observed (x) and predicted (y) TE$_{BC}$ with error bars representing the 25$^{th}$ and 75$^{th}$ percentile values derived from k-fold cross validation (k = 10) using stratified random sampling (Sect. 2.7). Ideal values are denoted by the red dashed lines such that a better predictor would fall closer to the intersection of the two lines. Top eight predictors per group (panel) are colored non-gray while the rest of the predictors are plotted in gray to show the relative performance of all predictors. Note that PERSIANN-CDR has been abbreviated to P-CDR (b-c). Panels share the same X- and Y-axis limits.**

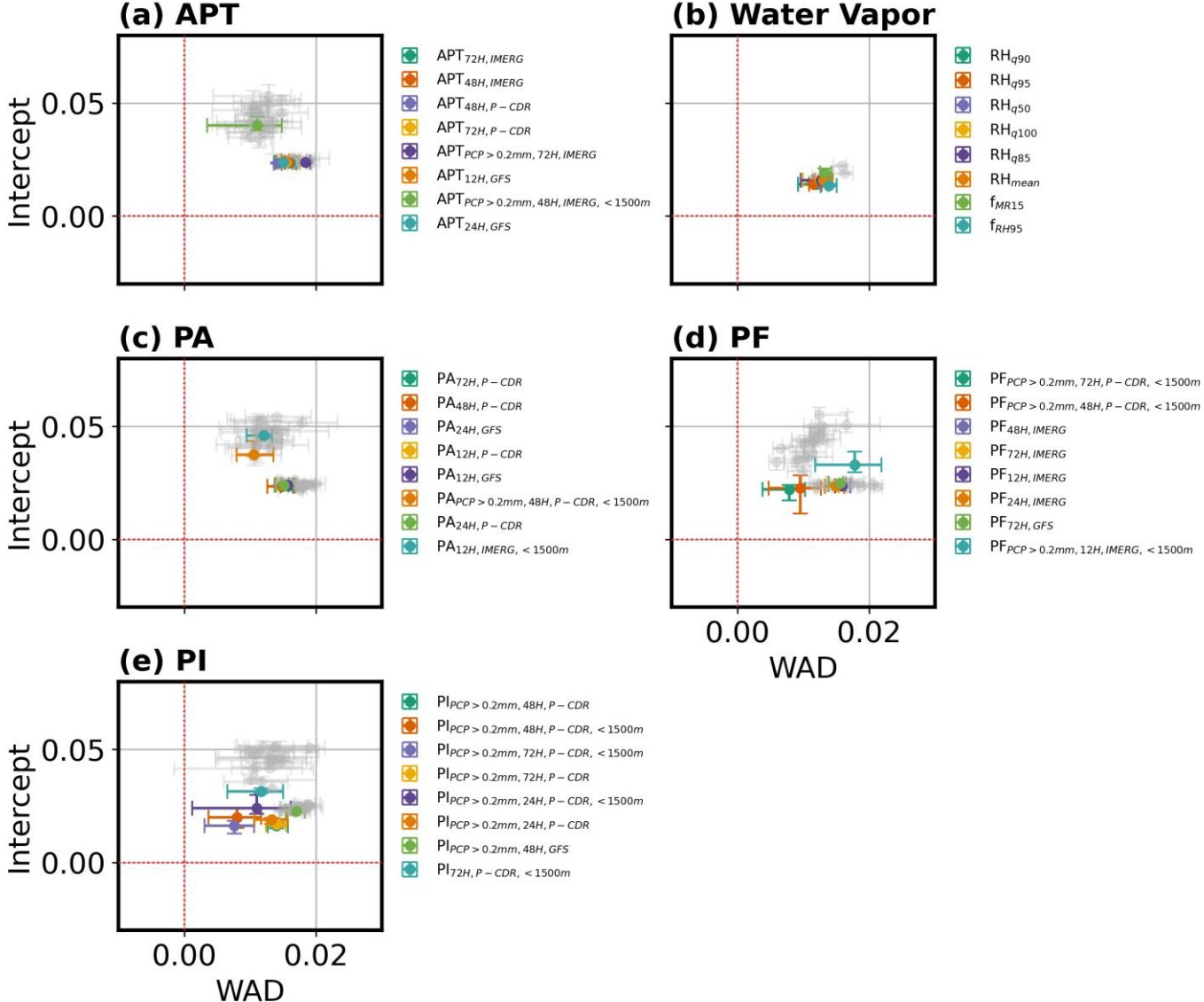

**Figure 3: Same as Fig. 2 but comparing intercept and weighted area difference (WAD, Sect. 2.7).**

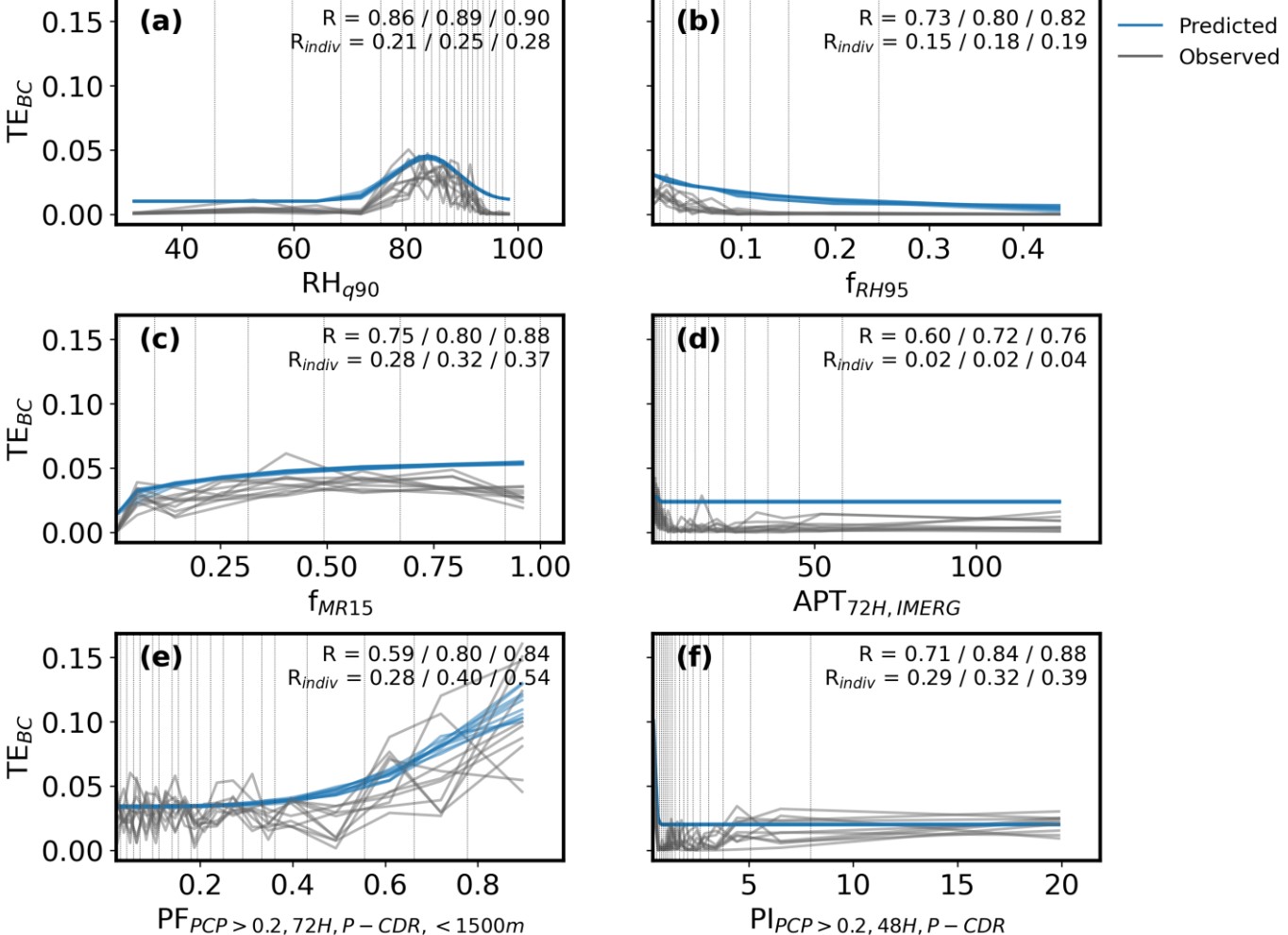

**Figure 4: Median values of observed (black) and predicted TE_BC (blue) as a function of selected predictors for 10 iterations during k-fold cross-validation. Pearson correlations are annotated as 25th/50th/75th percentiles from k-fold iterations (k=10) calculated in two ways: comparing predicted and observed median TE_BC per 5th percentile block of the predictor (R) and comparing predicted and observed TE_BC for individual points (R_indiv). Panels share the same Y-axis limits. Percentile blocks for each X-axis variable are denoted by vertical gray lines.**

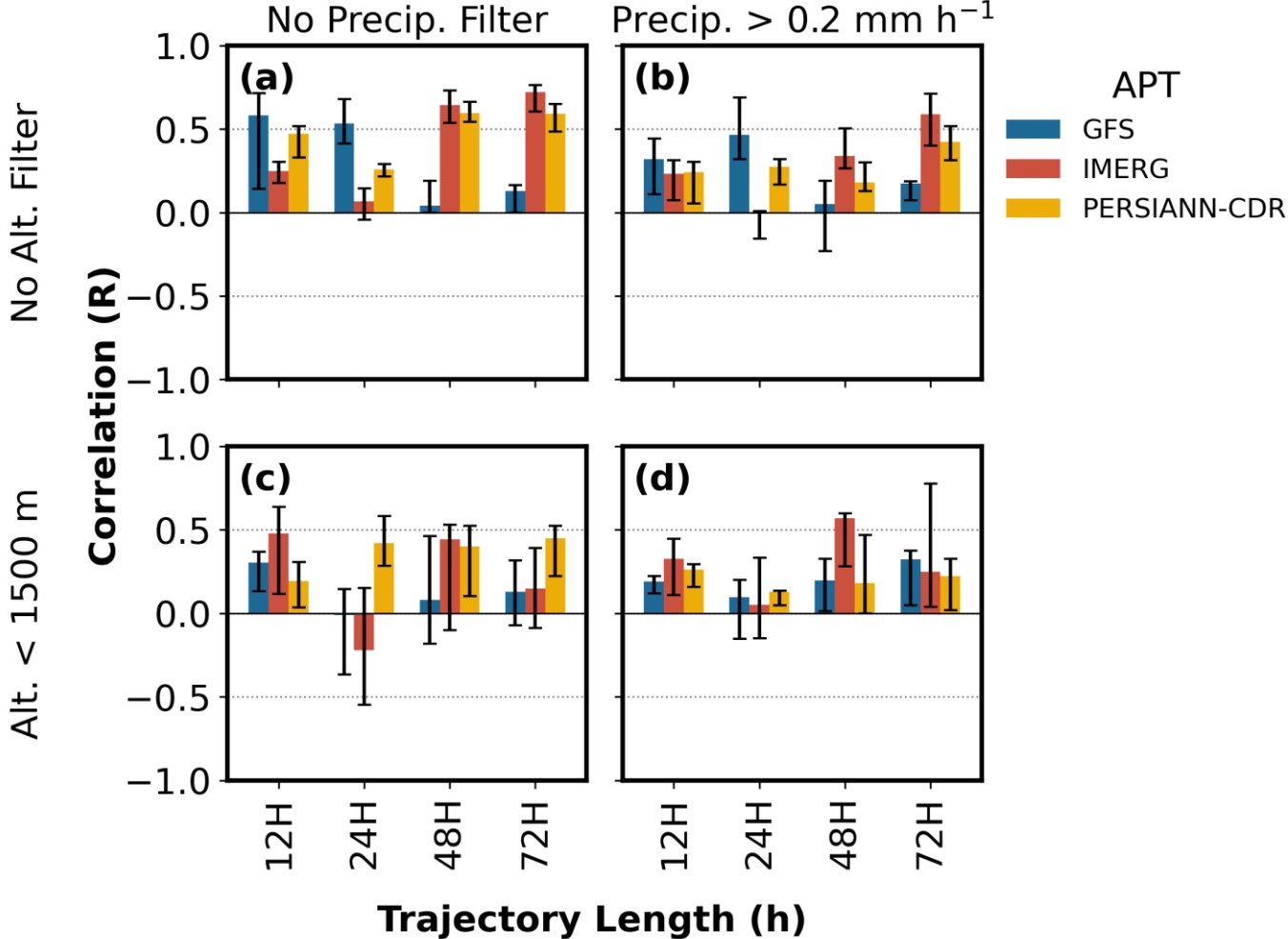

Figure 5: Pearson correlations (R) between observed TE$_{BC}$ and TE$_{BC}$ predicted by accumulated precipitation along trajectories (APT) for different trajectory lengths and precipitation data products. Each panel refers to a combination of altitude and precipitation intensity filters. Panels share the same X- and Y-axis limits.