# Peer review of "Assessing Potential Indicators of Aerosol Wet Scavenging During Long-Range Transport"

_EGUsphere, 2023_

## Referee Comment (RC1)

Comments on "**Identifying Better Indicators of Aerosol Wet Scavenging During Long-Range Transport**" by Hilario et al.

**General**

This paper describes the results from the data analyses of the wet scavenging of black carbon (BC) aerosols. The authors attempted to seek possible good indicators to describe the wet removal of BC during transport. In previous studies, the precipitation amount accumulated along backward trajectories (APT) has been analyzed as one of the indicators. Beside this, the authors suggested that the other several indicators related to the precipitation and humidity along the backward trajectories can well account for the variation of the degree of the removal of BC which is defined as the enhancement of BC to CO ($\Delta BC/\Delta CO$ ratio). The major results and discussion in this study meet the scope of Atmospheric Measurement Technology. Despite the significance of this study, there are several important issues to be addressed before accepting the manuscript. Please consider the following comments and necessary revisions of the data analyses and the descriptions in the manuscript.

**Major comments**

**1. Calculations of the enhancement ratio of BC to CO ($\Delta BC/\Delta CO$)**

The serious mistake is the choice of the enhancement ratio "$\Delta BC/\Delta CO$" for the quantitative investigations of wet scavenging of BC. This is because (1) the degree of the removal depends on the emission ratio of BC to CO ($ER_{BC2CO} \equiv \Delta BC/\Delta CO$ at the emission), (2) the background levels of BC and CO can vary with the air mass origins, and (3) $\Delta BC/\Delta CO$ can vary with the air mass mixing during the long-range transport. As to (1), authors also stated this point in section 4 (Limitations). Authors can take care of this point by analyzing the $ER_{BC2CO}$ from the observation data sets. In many previous studies using APT, this kind of adequate data preparations were conducted. For this purpose, the observed air masses need to be separated according to the air mass origins and/or emission sources as shown in Hilario et al. (2021), and then the variability of $ER_{BC2CO}$ (not $\Delta BC/\Delta CO$–APT relationship) during the CAMP²Ex campaign needed to be analyzed. The authors must justify the important and critical assumption that the variability is enough small among the air mass origins and/or emission sources to use $\Delta BC/\Delta CO$ as a unified indicator for the removal of BC. Separation of air mass types can lead to decrease the number of data to be analyzed, as discussed in section 4. However, if the authors do not show that the observation-based $ER_{BC2CO}$ did not largely vary among the different air mass origins/emission sources in

the study region, this is not a factor of the methodological limitations but it is just one of large error sources in the data analyses and the following interpretations. Please justify this assumption. If it were not for the validity, the data sets could not be suitable for the validation of authors' proposed method. As to (2), authors determined the background levels of BC and CO by analyzing the potential temperature profiles based on some previous studies. Currently, only the seasonal transition of background levels is considered by separating the periods to be analyzed. Are there any possibilities that the background levels vary depending on the air mass origins? As to (3), this effect depends on the time scale of the transport. The current manuscript is lacking in this information. Also regarding (1) and (2), it is needed to describe and discuss the observed feature of BC in the CAMP$^2$Ex campaign such as the relationship between the observed enhancements of BC and CO concentrations and backward trajectories (e.g., air mass origins) and typical transport time from the source regions. The former can affect the variabilities of $ER_{BC2CO}$. The latter can provide the insight into the adequate integration time for calculating the APT. Kanaya et al. (2016) indeed set 3 days for the integration time to calculate the APT by considering typical transport time from the source regions (e.g., central China) to the observation site (remote island in western Japan).

**2. The criteria to evaluate the performance of the predictors**

In section 2.5, authors stated "We use R because we are more interested in ~". The performance of the combination of the predictors and fitted functions was evaluated by comparing the Pearson correlation coefficients (R) of the correlations to account for the variations of the observed $\Delta BC/\Delta CO$. Therefore, the accuracy of the predication (i.e., slope and WAD) was not weighted in this study, resulting in the inaccurate performance of almost all the predictions that seriously overestimate the observed values of $\Delta BC/\Delta CO$ especially for their low value ranges (positive values of the WAD and intercepts). To me as a potential reader of this paper, this fact suggests that the approach proposed in this study is not always better than the previous works. The APT approach used in the previous studies showed the better performance to predict the $\Delta BC/\Delta CO$ or transport efficiency ($TE_{BC} = (\Delta BC/\Delta CO)/ER_{BC2CO}$) using the long-term averaged data sets (e.g., Kanaya et al., 2016; Choi et al., 2020). Although the correlation of the $TE_{BC}$ and APT was not so good, based on the binned average data sets of $TE_{BC}$-APT relationship, the decreasing tendency of the $TE_{BC}$ during the transport was successfully predicted by the APT in their studies. In this study, the APT-based prediction skills were not fully described. The predictions with the lower R should be

discussed for the fair and comprehensive evaluations of the accuracy of all the predictions tried in this study.

**3. Calculating the APT and other indicators**

Authors might misunderstand the previous studies to apply the APT in their data analyses. As an example of ground-based studies, Kanaya et al. (2016) defined the length of the total backward time to calculate the trajectories (5 days) and integration time to calculate the APT (3 days) by considering the typical source areas (East Asian continent) affecting the observation site (a remote island in western Japan) and the typical meteorological field. Oshima et al. (2012; 2013) analyzed the aircraft observation data sets of BC and CO for uplifted air parcels sampled at the upper atmosphere (3–6 km), and investigated the effect of upward transport of air masses associated with the precipitation. The APT for uplifted air masses were calculated by integrating the precipitation water content from the uplifted location to the sampling point (Oshima et al., 2012). Depending on the definitions of the APT, the sensitivity of the precipitation to the transport efficiency of BC was significantly different from those from ground-based investigations (e.g., Kanaya et al., 2016). What I would like to claim is that possible indicators for the wet removals of BC should be designed by the careful consideration of the actual atmospheric conditions (i.e., meteorology) and the observation types (e.g., ground vs. aircraft).

In this study, the basic characterizations of how the air parcels sampled at the aircraft observatory were transported from where (i.e., transport pathway), and the atmospheric transport time scale from the possible source area/region of BC and CO are critically missing (Referring Hilario et al. (2021) in section 2.1 is insufficient.). In section 3 "Results and discussion part", author should prepare additional subsection to describe the observed features of BC aerosols during CAMP$^2$Ex campaign to clarify the above points (This was also pointed out in the comment 1). Based on the descriptions about the basic data analyses of the BC and CO enhancements, authors should define the proper length of the backward calculations of the trajectories and integration time for calculating various parameters in relation to the removal of BC.

**4. Curve-fitting equations**

The authors prepared 4 equations in the data analyses. Two of them (Oshima and Kanaya (stretched exponential)) were derived from the previous studies analyzing the TE$_{BC}$–APT relationship. What was the basis to apply the remaining two? In this study, four types of parameters other than APT were analyzed, however two types of the

equations were applied to these.   Are there needs to apply and test more equations to these various indicators?   Please clarify the reason why the authors decide to select these two equations for the non-APT parameters.

**5.  How should we judge as "Better" when the performance to predict the removal of BC from atmosphere is evaluated?**

In relation to the above comments, I strongly suggest not to use "identifying" and "better" in the title of the current manuscript.   At least, the proposed approaches are not better than the previous works using the APT.   So, the better indicators were not identified yet.   More careful discussion based on more careful data analyses is needed to justify it is "better".   Please consider the significant revisions of the data analyses and descriptions in the manuscript.

**Minor comments**

**P5, L146.** Please clarify how authors determined the number of k for k-fold cross validation analyses.

**P5, L147.** "jug" should be "Jug". What is the version of Jug used in this study?

**P7, L232–L234.** "We hypothesize ~" High RH condition is also related to in-cloud condition as well as precipitation as suggested.   It is well known that BC can be activated to form cloud droplets as CCN.   This results in the removal of BC from atmosphere (into hydrometeor).   Needs to be revised accordingly.

**P20, Table 2.** The curve-fitting equation which produces the highest value of R should be added with each predictor listed in Table 2.   This will help us to easily find which equation works well with which indicator.   Adding typical values of the coefficients in the fitted curves to the list is highly recommended for the clarity of the shape of the determined curves.

**P22, Figure 1b.** To clarify the percentile blocks, please consider to modify the figure style of the lines between points to the lines between markers especially for the predicted traces.

**P25, Figure 4.** Same as the comment to Figure 1b.

**References**

Choi, Y., Kanaya, Y., Takigawa, M., Zhu, C., Park, S.-M., Matsuki, A., Sadanaga, Y., Kim, S.-W., Pan, X., and Pisso, I.: Investigation of the wet removal rate of black carbon in East Asia: validation of a below- and in-cloud wet removal scheme in FLEXiblePARTicle (FLEXPART) model v10.4, Atmos. Chem. Phys., 20, 13655–13670, https://doi.org/10.5194/acp-20-13655-2020, 2020.

Hilario, M. R. A., Crosbie, E., Shook, M., Reid, J. S., Cambaliza, M. O. L., Simpas, J. B. B., Ziemba, L., DiGangi, J. P., Diskin, G. S., Nguyen, P., Turk, F. J., Winstead, E., Robinson, C. E., Wang, J., Zhang, J., Wang, Y., Yoon, S., Flynn, J., Alvarez, S. L., Behrangi, A., and Sorooshian, A.: Measurement report: Long-range transport patterns into the tropical northwest Pacific during the CAMP$^2$Ex aircraft campaign: chemical composition, size distributions, and the impact of convection, Atmos. Chem. Phys., 21, 3777–3802, https://doi.org/10.5194/acp-21-3777-2021, 2021.

Kanaya, Y., Pan, X., Miyakawa, T., Komazaki, Y., Taketani, F., Uno, I., and Kondo, Y.: Long-term observations of black carbon mass concentrations at Fukue Island, western Japan, during 2009?2015: constraining wet removal rates and emission strengths from East Asia, Atmos. Chem. Phys., 16, 10689–10705, https://doi.org/10.5194/acp-16-10689-2016, 2016.

Oshima, N., Kondo, Y., Moteki, N., Takegawa, N., Koike, M., Kita, K., Matsui, H., Kajino, M., Nakamura, H., Jung, J.S., and Kim, Y.J.: Wet removal of black carbon in Asian outflow: Aerosol Radiative Forcing in East Asia (A-FORCE) aircraft campaign, J. Geophys. Res., 117, D03204, doi:10.1029/2011JD016552, 2012.

Oshima, N., Koike, M., Kondo, Y., Nakamura, H., Moteki, N., Matsui, H., Takegawa, N., and Kita, K.: Vertical transport mechanisms of black carbon over East Asia in spring during the A-FORCE aircraft campaign, J. Geophys. Res.-Atmos., 118, 13175–13198, https://doi.org/10.1002/2013JD020262, 2013.

---

## Author Comment (AC1)

Manuscript ID: egusphere-2023-726
TITLE: Assessing Potential Indicators of Aerosol Wet Scavenging During Long-Range Transport

We thank the handling editor and the two referees for their helpful comments. We have provided our responses to reviewer comments below in blue. One major change we want to mention is that we now set the transport efficiency of black carbon ($TE_{BC}$) as the predictand of our method rather than the enhancement ratio of black carbon and carbon monoxide ($\Delta BC/\Delta CO$) in response to major comments from both reviewers.

Comments on "**Identifying Better Indicators of Aerosol Wet Scavenging During Long-Range Transport**" by Hilario et al.

**General**

This paper describes the results from the data analyses of the wet scavenging of black carbon (BC) aerosols. The authors attempted to seek possible good indicators to describe the wet removal of BC during transport. In previous studies, the precipitation amount accumulated along backward trajectories (APT) has been analyzed as one of the indicators. Beside this, the authors suggested that the other several indicators related to the precipitation and humidity along the backward trajectories can well account for the variation of the degree of the removal of BC which is defined as the enhancement of BC to CO ($\Delta BC/\Delta CO$ ratio). The major results and discussion in this study meet the scope of Atmospheric Measurement Technology. Despite the significance of this study, there are several important issues to be addressed before accepting the manuscript. Please consider the following comments and necessary revisions of the data analyses and the descriptions in the manuscript.

**Major comments**

**1. Calculations of the enhancement ratio of BC to CO ($\Delta BC/\Delta CO$)**

The serious mistake is the choice of the enhancement ratio "$\Delta BC/\Delta CO$" for the quantitative investigations of wet scavenging of BC. This is because (1) the degree of the removal depends on the emission ratio of BC to CO ($ER_{BC2CO} \equiv \Delta BC/\Delta CO$ at the emission), (2) the background levels of BC and CO can vary with the air mass origins, and (3) $\Delta BC/\Delta CO$ can vary with the air mass mixing during the long-range transport. As to (1), authors also stated this point in section 4 (Limitations). Authors can take care of this point by analyzing the $ER_{BC2CO}$ from the observation data sets. In many previous studies using APT, this kind of adequate data preparations were conducted. For this purpose, the observed air masses need to be separated according to the air mass origins and/or emission sources as shown in Hilario et al. (2021), and then the variability of $ER_{BC2CO}$ (not $\Delta BC/\Delta CO$–APT relationship) during the CAMP²Ex campaign needed to be analyzed. The authors must justify the important and critical assumption that the variability is enough small among the air mass origins and/or emission sources to use $\Delta BC/\Delta CO$ as a unified indicator for the removal of BC. Separation of air mass types can lead to decrease the number of data to be analyzed, as discussed in section 4. However, if the authors do not show that the observation-based $ER_{BC2CO}$ did not largely vary among the different air mass origins/emission sources in the study region, this is not a factor of the methodological limitations but it is just one of large error sources in the data analyses and the following interpretations. Please justify this assumption. If it were not for the validity, the data sets could not be suitable for the validation of authors' proposed method.

Response: Thank you for the suggestions. We have taken your suggestion and incorporated emission values of BC and CO from the Copernicus Atmosphere Monitoring Service (CAMS) Global Anthropogenic Emissions (CAMS-GLOB-ANT) version 5.3 (Soulie et al., 2023) which has a spatial resolution of 0.1x0.1° and a temporal resolution of one month. Emissions from CAMS-GLOB-ANT account for 17 emission sectors, including shipping, and are

reported in units of mass flux (kg m⁻² s⁻¹). These details have been added to the Methods section (Sect 2.4).

To combine the emission fluxes with the HYSPLIT backward trajectories, we calculated the mean BC/CO emission ratio ($ER_{BC/CO}$) along each 72-h trajectory, inverse-weighted by altitude. The weighting function (below, left) reflects the higher likelihood of surface influence when the trajectory is at a lower altitude. An example of the weighting function as a function of trajectory altitude is shown below (right), wherein the trajectory ascends from 1 km to 6 km and weights decrease in response to higher altitudes. As a result, the trajectory's mean $ER_{BC/CO}$ will be determined mainly by timesteps when the trajectory is close to the surface. We note that we did not restrict the calculation of $ER_{BC/CO}$ to source regions but included all points along the trajectories such that $ER_{BC/CO}$ in this study represents the weighted-average emission ratio of BC and CO encountered by the transported air mass over the past 72 hours.

[Figure]

**Figure S2: (a) Weighting function used when calculating average emission ratios along trajectories. (b) An example of the inverse relationship between trajectory altitude (left y-axis, blue) and assigned weight (right y-axis, orange). The x-axis of (b) is trajectory timestep where a timestep of zero is when the trajectory reaches the aircraft.**

We can then quantify the transport efficiency of BC (TE_BC) using Eq. 1:

$$TE = \frac{\left(\frac{\Delta BC}{\Delta CO}\right)_{receptor}}{ER_{BC/CO}} \qquad (1)$$

In our original submission, $\left(\frac{\Delta BC}{\Delta CO}\right)_{receptor}$ was in units of μg m⁻³ ppmv⁻¹. To ensure TE_BC would be unitless, we converted $\Delta CO$ from ppmv to μg m⁻³ using ambient pressure and temperature measured by the aircraft (added to Sect. 2.2). We note that $ER_{BC/CO}$ is not required to be an enhancement ratio. The enhancement calculation is only necessary at the receptor site (i.e., $\left(\frac{\Delta BC}{\Delta CO}\right)_{receptor}$) to account for the local background concentrations over the tropical West Pacific.

The TE$_{BC}$ and $\Delta$BC/$\Delta$CO are strongly correlated (R$^2$ = 0.90), suggesting that the updated results from using TE$_{BC}$ as the predictand will be quite similar to our prior results when we used $\Delta$BC/$\Delta$CO as the predictand. This suggests that the major conclusions of the paper will be largely unchanged. Details of the emission inventory and its incorporation into our calculation of emission ratios have been added to Sect. 2.4-2.5.

To show the variation of $ER_{BC/CO}$ with source region, we provide below $ER_{BC/CO}$ (Fig. S1c) and TE$_{BC}$ (Fig. S1d) resolved by source region (East Asia (EA); Maritime Continent (MC); Peninsular Southeast Asia (PSEA); West Pacific (WP)). The source identification was performed in Hilario et al. (2021) which classified backward trajectories into source regions using bounding boxes over major source regions established in previous literature. In addition to passing over source region bounding boxes, the source classification also considered (1) trajectory altitude, specifically whether or not the trajectory was below 2 km AGL which conservatively approximates climatological boundary layer heights over the region (Chien et al., 2019), as well as (2) trajectory residence time within each bounding box (minimum residence time: 6 hours). These details are now included in Sect 2.2.

The resulting source-resolved distributions of the BC/CO emission ratio and TE$_{BC}$ show some variation by source region; however, ER$_{BC/CO}$ across source regions (Fig. S1c) generally falls within 0.23 - 0.26. The resulting TE$_{BC}$ values (Fig. S1d) are generally below 0.1, which mean less than 10% of the emitted BC/CO reaches the receptor site. The similarities between ER$_{BC/CO}$ per source region may explain the strong correlation between TE$_{BC}$ and $\Delta$BC/$\Delta$CO (R$^2$ = 0.90).

[Figure]

**Figure S1: (a) The enhancement ratio of black carbon (BC) to carbon monoxide (CO) (ΔBC/ΔCO; unitless) per source region (EA: East Asia, MC: Maritime Continent, PSEA: Peninsular Southeast Asia, WP: West Pacific; identified in Hilario et al. (2021)), (b) transit times (in hours) from major source regions, (c) emission ratios of BC/CO ($ER_{BC/CO}$) along each trajectory using the CAMS-GLOB-ANT emissions (Sect. 2.4-2.5), and (d) transport efficiencies of BC ($TE_{BC}$). Boxplots for WP in (a) and (d) were removed due to a low number of data points (N = 2) remaining after the ΔCO > 0.02 ppm filter (Sect. 2.2).**

As to (2), authors determined the background levels of BC and CO by analyzing the potential temperature profiles based on some previous studies. Currently, only the seasonal transition of background levels is considered by separating the periods to be analyzed. Are there any possibilities that the background levels vary depending on the air mass origins?

Response: As we aim to study scavenging during transport, the background refers to local concentrations of BC and CO assuming no transport event is taking place. Because of this, the background levels of a receptor site do not vary with air mass origin. In our response to your comment on considering source regions, we have adjusted our analysis to be source-resolved and focusing on predicting the transport efficiency of BC instead of ΔBC/ΔCO.

As to (3) , this effect depends on the time scale of the transport. The current manuscript is lacking in this information.

Response: We have added this information to the Limitations section:

"An underlying assumption is that there is negligible entrainment of additional BC and CO during transport. Dilution via mixing during transport is also expected to influence the ΔBC/ΔCO ratio. Thus, our analysis assumes that wet scavenging is the main driver of changes in ΔBC/ΔCO and chemical transport modelling in future work is needed to quantify the effect of mixing on ΔBC/ΔCO."

Also regarding (1) and (2), it is needed to describe and discuss the observed feature of BC in the CAMP²Ex campaign such as the relationship between the observed enhancements of BC and CO concentrations and backward trajectories (e.g., air mass origins) and typical transport time from the source regions. The former can affect the variabilities of $ER_{BC2CO}$. The latter can provide the insight into the adequate integration time for calculating the APT. Kanaya et al. (2016) indeed set 3 days for the integration time to calculate the APT by considering typical transport time from the source regions (e.g., central China) to the observation site (remote island in western Japan).

Response: We have calculated and plotted transport times from source regions and included it as Fig. S1b (provided above in response to major comment #1). We note that the application of our method was done on all CAMP²Ex data, not just data where a source region was identified. Generally, the distribution of transit times across all source regions shows values below 72 hours, suggesting that 72 hours is sufficient to capture long-range transport from major source regions. We added this description to Sect. 2.3 of the paper.

**2.  The criteria to evaluate the performance of the predictors**

In section 2.5, authors stated "We use R because we are more interested in ~". The performance of the combination of the predictors and fitted functions was evaluated by comparing the Pearson correlation coefficients (R) of the correlations to account for the variations of the observed ΔBC/ΔCO. Therefore, the accuracy of the predication (i.e., slope and WAD) was not weighted in this study, resulting in the inaccurate performance of almost all the predictions that seriously overestimate the observed values of ΔBC/ΔCO especially for their low value ranges (positive values of the WAD and intercepts). To me as a potential reader of this paper, this fact suggests that the approach proposed in this study is not always better than the previous works. The APT approach used in the previous studies showed the better performance to predict the ΔBC/ΔCO or transport efficiency ($TE_{BC}$ = (ΔBC/ΔCO)/$ER_{BC2CO}$) using the long-term averaged data sets (e.g., Kanaya et al., 2016; Choi et al., 2020).

Although the correlation of the TE$_{BC}$ and APT was not so good, based on the binned average data sets of TE$_{BC}$-APT relationship, the decreasing tendency of the TE$_{BC}$ during the transport was successfully predicted by the APT in their studies. In this study, the APT-based prediction skills were not fully described. The predictions with the lower R should be discussed for the fair and comprehensive evaluations of the accuracy of all the predictions tried in this study.

Response: Thank you for the suggestions. To clarify what we meant by "We use R because we are more interested in~", we performed curve-fitting using the different equations in Table 2 and selected the equation that gave the best R between predicted and observed TE$_{BC}$. The comparison of different predictors in Sect. 3 considers all statistical metrics such as WAD and intercept. The new text reads:

"When selecting which equation to use (among those in Table 2) for fitting between the predictor and TE$_{BC}$, we opted for the equation that resulted in the highest R between observed and predicted TE$_{BC}$ (e.g., Fig. 1c). The basis of this choice on R was because our objective is to identify predictors that can at least capture trends in TE$_{BC}$. After selecting which equation to use per predictor, the subsequent comparison (Sect. 3) of the performance of different predictors considers other statistical metrics such as slope, intercept, and WAD."

To show the relative performance of all predictors (not just those with the best statistical metrics), we have edited Figs. 2, 3, S3 to show all predictors including those that had lower R (in grey). The updated Fig. 2 is provided below. To simplify the plot while still showing all predictors, we colored only the top eight predictors with highest R per panel. The new figures show the difference in performance across predictors and the comparatively better performance of predictors we discussed in the previous iteration of the draft. We have added more discussion to the text to consider multiple statistical metrics when assessing performance (slope, WAD, intercept, etc.).

We have added more discussion on the APT-based prediction skills in Sect. 3.1 and 3.3. However, as we have a total of 204 variables that we tested in our study, a comprehensive discussion of all 204 predictors is not possible which is why we focused our discussion on the best-performing predictors.

[Figure]

**Figure 2: Slope and Pearson correlation (R) values derived from linear regressions of observed (x) and predicted (y) TE$_{BC}$ with error bars representing the 25$^{th}$ and 75$^{th}$ percentile values derived from k-fold cross validation (k = 10) using stratified random sampling (Sect. 2.5). Ideal values are denoted by the red dashed lines such that a better predictor would fall closer to the intersection of the two lines. Top eight predictors per group (panel) are colored non-grey while the rest of the predictors are plotted in grey to show the relative performance of all predictors. Note that PERSIANN-CDR has been abbreviated to P-CDR (b-c). Panels share the same X- and Y-axis limits.**

**3. Calculating the APT and other indicators**

Authors might misunderstand the previous studies to apply the APT in their data analyses. As an example of ground-based studies, Kanaya et al. (2016) defined the length of the total backward time to calculate the trajectories (5 days) and integration time to calculate the APT (3 days) by considering the typical source areas (East Asian continent) affecting the observation site (a remote island in western Japan) and the typical meteorological field. Oshima et al. (2012; 2013) analyzed the aircraft observation data sets of BC and CO for uplifted air parcels sampled at the upper atmosphere (3–6 km), and investigated the effect of upward transport of air masses associated with the precipitation. The APT for uplifted air masses were calculated by integrating the precipitation water content from the uplifted location to the sampling point (Oshima et al., 2012). Depending on the definitions of the APT, the sensitivity of the precipitation to the transport efficiency of BC was significantly different from those from ground-based investigations (e.g., Kanaya et al., 2016). What I would like to claim is that possible indicators for the wet removals of BC should be designed by the careful consideration of the actual atmospheric conditions (i.e., meteorology) and the observation types (e.g., ground vs. aircraft).

In this study, the basic characterizations of how the air parcels sampled at the aircraft observatory were transported from where (i.e., transport pathway), and the atmospheric transport time scale from the possible source area/region of BC and CO are critically missing (Referring Hilario et al. (2021) in section 2.1 is insufficient.). In section 3 "Results and discussion part", author should prepare additional subsection to describe the observed features of BC aerosols during CAMP$^2$Ex campaign to clarify the above points (This was also pointed out in the comment 1). Based on the descriptions about the basic data analyses of the BC and CO enhancements, authors should define

the proper length of the backward calculations of the trajectories and integration time for calculating various parameters in relation to the removal of BC.

Response: The objective of our study is to demonstrate a method for identifying potential predictors of wet scavenging. A characterization of transport patterns during the CAMP$^2$Ex field campaign was done in our past study (Hilario et al., 2021) which included CO and BC. To provide information on the basic characteristics of BC and CO during CAMP$^2$Ex, we have provided a description of the transport of BC and CO in Appendix A. To justify the choice of the 72-hour backward trajectories, Fig. S1b shows the distribution of transport times from different source regions (Sect. 2.2) to the CAMP$^2$Ex aircraft. Generally, transit times are below 72 hours suggesting that 72 hours is sufficient to capture long-range transport from major source regions into the tropical West Pacific. We added this information on transit times in Sect. 2.3.

**4. Curve-fitting equations**

The authors prepared 4 equations in the data analyses. Two of them (Oshima and Kanaya (stretched exponential)) were derived from the previous studies analyzing the TEBC–APT relationship. What was the basis to apply the remaining two? In this study, four types of parameters other than APT were analyzed, however two types of the equations were applied to these. Are there needs to apply and test more equations to these various indicators? Please clarify the reason why the authors decide to select these two equations for the non-APT parameters.

Response: The two equations (Gaussian, general exponential) were selected as they represent typical statistical distributions. For some predictors (e.g., precipitation), we expected a general exponential relationship wherein higher precipitation leads to exponentially lower $TE_{BC}$. For $RH_q$, its relationship with $TE_{BC}$ appeared Gaussian where $TE_{BC}$ was highest at around RH = 85% (Fig. 1b). As none of the other equations in Table 1 captured this relationship well, we opted to include a Gaussian function in our equations. We note that, as we aim to find good meteorological predictors of $TE_{BC}$, these predictors do not necessarily need to have an inversely proportional relationship with $TE_{BC}$. Thus, the Gaussian and general exponential functions were included to consider a wider range of potential predictors of $TE_{BC}$. We have added this explanation to the text in Sect. 2.5 and Sect. 4.

**5. How should we judge as "Better" when the performance to predict the removal of BC from atmosphere is evaluated?**

In relation to the above comments, I strongly suggest not to use "identifying" and "better" in the title of the current manuscript. At least, the proposed approaches are not better than the previous works using the APT. So, the better indicators were not identified yet. More careful discussion based on more careful data analyses is needed to justify it is "better". Please consider the significant revisions of the data analyses and descriptions in the manuscript.

Response: The updated title now reads "Assessing Potential Indicators of Aerosol Wet Scavenging During Long-Range Transport". In response to your major comment #1, we have updated the method to predict the transport efficiency of black carbon ($TE_{BC}$) instead of $\Delta BC/\Delta CO$ by incorporating emission ratios from the CAM-GLOB-ANT inventory (described in Sect. 2.4). By accounting for emissions along the trajectory (described in Sect. 2.5), this new method focused on $TE_{BC}$ has reduced uncertainties regarding the confounding influence of source emissions during transport. We have updated our figures and writing accordingly.

We have also updated the text to explain more clearly why we wrote that some predictors performed better than others by mentioning statistical metrics (e.g., slope, R) derived using the regression, curve-fitting, and k-fold crossvalidation procedure. Examples:

"Predictors based on specific quantiles of RH (e.g., $RH_{q90}$) also performed quite well in predicting both $TE_{BC}$ trends and magnitudes (intercepts close to zero, WAD close to zero, slopes close to 1, R close to 1)"

"Frequency-related predictors such as $f_{MR15}$ and $f_{RH95}$ performed better than APT in predicting $TE_{BC}$ trends (higher R) and magnitudes (slopes closer to 1)."

**Minor comments**

**P5, L146.** Please clarify how authors determined the number of k for k-fold cross validation analyses.

Response: The decision of what value of k to use relates to the bias-variance tradeoff (Hastie et al., 2009). A low value of k such as 5 would lead to cross-validation results (e.g., $R^2$ between observed and predicted $TE_{BC}$) that have low variance but potentially high bias due to the splitting of the dataset into five parts wherein 80% of the data are used for the training set and 20% are used for the testing set. For the opposite case (e.g., k = 20), the cross-validation results may have low bias but high variance. Previous work evaluating different accuracy estimation methods showed that k = 10 is sufficient to estimate performance metrics (e.g., $R^2$) while minimizing computational expense (Breiman and Spector, 1992; Kohavi, 1995). We have added this information to Sect. 2.7 to explain our choice of k.

We performed sensitivity testing by repeating our k-fold cross validation for different values of k (5, 10, 15, 20) and saw that there were minimal changes between k = 10 (Fig. 2) and other k values (k = 5 and 20 are provided below). We have added a summary of our sensitivity tests to the text in Sect 2.7, which reads:

"Sensitivity testing with the k value showed no significant effect on the general conclusions of the study when k was changed between 5 and 20 (not shown). We opted for k = 10 based on previous work evaluating different accuracy estimation methods which showed that k = 10 is sufficient to estimate performance metrics (e.g., R2) while minimizing computational expense (Breiman and Spector, 1992; Kohavi, 1995)."

Same as Fig. 2 but for k = 5:

[Figure]

Same as Fig. 2 but for k = 20:

[Figure]

**P5, L147.** "jug" should be "Jug". What is the version of Jug used in this study?

Response: jug has been updated to Jug. Version number (2.2.2) has been added to the text.

**P7, L232–L234.** "We hypothesize ~" High RH condition is also related to in-cloud condition as well as precipitation as suggested. It is well known that BC can be activated to form cloud droplets as CCN. This results in the removal of BC from atmosphere (into hydrometeor). Needs to be revised accordingly.

Response: In our explanation of the high RH, we have updated the text to:

"This possibility is supported by previous literature showing the tendency of SPPs to misestimate light (Nadeem et al., 2022; Kidd et al., 2021) or intense precipitation (Chen et al., 2020a; Gupta et al., 2020); however, we cannot rule out the possibility of the in-cloud activation (high RH) and subsequent removal of BC during transport. Thus, our hypothesis of RH from reanalysis capturing missed precipitation from SPPs requires further investigation in future work."

**P20, Table 2.** The curve-fitting equation which produces the highest value of R should be added with each predictor listed in Table 2. This will help us to easily find which equation works well with which indicator. Adding typical values of the coefficients in the fitted curves to the list is highly recommended for the clarity of the shape of the determined curves.

Response: We have added a table in the SI (Table S1) that shows the top 8 predictors for each panel in Fig. 2 (40 predictors total) as well as their equation and coefficients. We reference Table S1 in Sect. 3.1 to provide readers with this information.

**P22, Figure 1b.** To clarify the percentile blocks, please consider to modify the figure style of the lines between points to the lines between markers especially for the predicted traces.

Response: We have updated Fig. 1b to show the different percentile blocks used during curve-fitting:

[Figure]

**P25, Figure 4.** Same as the comment to Figure 1b.

Response: We have updated Fig. 4:

[Figure]

**Figure 4: Median values of observed (black) and predicted TE_BC (blue) as a function of selected predictors for 10 iterations during k-fold cross-validation. Pearson correlations are annotated as 25th/50th/75th percentiles from k-fold iterations (k=10) calculated in two ways: comparing predicted and observed median TE_BC per 5th percentile block of the predictor (R) and comparing predicted and observed TE_BC for individual points (R_indiv). Panels share the same Y-axis limits. Percentile blocks for each X-axis variable are denoted by vertical grey lines.**

**References**

Choi, Y., Kanaya, Y., Takigawa, M., Zhu, C., Park, S.-M., Matsuki, A., Sadanaga, Y., Kim, S.-W., Pan, X., and Pisso, I.: Investigation of the wet removal rate of black carbon in East Asia: validation of a below- and in-cloud wet removal scheme in FLEXiblePARTicle (FLEXPART) model v10.4, Atmos. Chem. Phys., 20, 13655– 13670, https://doi.org/10.5194/acp-20-13655-2020, 2020.

Hilario, M. R. A., Crosbie, E., Shook, M., Reid, J. S., Cambaliza, M. O. L., Simpas, J. B., Ziemba, L., DiGangi, J. P., Diskin, G. S., Nguyen, P., Turk, F. J., Winstead, E., Robinson, C. E., Wang, J., Zhang, J., Wang, Y., Yoon, S., Flynn, J., Alvarez, S. L., Behrangi, A., and Sorooshian, A.: Measurement report: Long-range transport patterns into the tropical northwest Pacific during the CAMP²Ex aircraft campaign: chemical composition, size distributions, and the impact of convection, Atmos. Chem. Phys., 21, 3777–3802, https://doi.org/10.5194/acp-21-3777-2021, 2021.

Kanaya, Y., Pan, X., Miyakawa, T., Komazaki, Y., Taketani, F., Uno, I., and Kondo, Y.: Long-term observations of black carbon mass concentrations at Fukue Island, western Japan, during 2009?2015: constraining wet removal rates and emission strengths from East Asia, Atmos. Chem. Phys., 16, 10689–10705,

https://doi.org/10.5194/acp-16-10689-2016, 2016.

Oshima, N., Kondo, Y., Moteki, N., Takegawa, N., Koike, M., Kita, K., Matsui, H., Kajino, M., Nakamura, H., Jung, J.S., and Kim, Y.J.: Wet removal of black carbon in Asian outflow: Aerosol Radiative Forcing in East Asia (A-FORCE) aircraft campaign, J. Geophys. Res., 117, D03204, doi:10.1029/2011JD016552, 2012.

Oshima, N., Koike, M., Kondo, Y., Nakamura, H., Moteki, N., Matsui, H., Takegawa, N., and Kita, K.: Vertical transport mechanisms of black carbon over East Asia in spring during the A-FORCE aircraft campaign, J. Geophys. Res.-Atmos., 118, 13175–13198, https://doi.org/10.1002/2013JD020262, 2013.

Referee 2:

This study aims to identify meteorological variables affecting precipitation scavenging of atmospheric aerosols using a combination of aircraft, satellite, and reanalysis data augmented by trajectory modeling to account for air mass history. In literature, key variables controlling below-cloud aerosol scavenging have been well documented, although existing parametrizations still have large uncertainties (especially for particles in the submicron size range). This study addresses the former (key variables), but added little knowledge on the latter (how to reduce the uncertainties in parametrizations). The analysis approach is also subject to large uncertainties. I only provided a few comments related to the science of precipitation scavenging for the authors to consider in improving the quality of the manuscript.

Lines 25-28: Most existing parameterizations for precipitation scavenging of atmospheric aerosols have considered precipitation intensity, precipitation amount, and raindrop and aerosol size distributions, etc. It is not clear what (additional) variables this study proposes (after reading the whole abstract) for better parametrizing wet scavenging of atmospheric aerosols. The authors are encouraged to provide an explicit recommendation instead of a general statement. This comment also applies to the Conclusions section (the last paragraph) since no clear message of their recommendations is provided.

Response: We have reworded the conclusions and abstract to be clearer on our recommendations given our findings.

Section 2.2: If BC and CO have the same sources but different sinks (one is wet scavenged and the other is not), then using this ratio approach is reasonable in tracking the wet scavenging of BC. However, if they have different sources along the air mass trajectory (which is likely the case), this approach would cause very larger uncertainties.

Response: The BC to CO ratio will indeed vary by source and was an important uncertainty in our original method. In our original method, $\Delta BC/\Delta CO$ is assumed to be influenced by two main factors: (1) source emissions of BC and CO along the trajectory path, and (2) removal of BC via wet scavenging. In response to one of Reviewer #1's major comments, we have updated our method with $TE_{BC}$ as the predictand instead of $\Delta BC/\Delta CO$. $TE_{BC}$ is defined in Eq. (1) (Sect. 2.5) and involves the mean BC/CO emission ratio ($ER_{BC/CO}$) along each trajectory. $ER_{BC/CO}$ represents the weighted-average surface emission ratio of BC and CO encountered by the transported air mass over the past 72 hours. To calculate $ER_{BC/CO}$, we use emissions from the CAMS-GLOB-ANT emission inventory (Sect. 2.4) and calculate $ER_{BC/CO}$ that is inverse-weighted by trajectory altitude to reflect the higher likelihood of surface influence when trajectories are close to the surface (Sect. 2.5). By using $TE_{BC}$ that accounts for $ER_{BC/CO}$, we reduce uncertainties from sources along the trajectory path such that $TE_{BC}$ is reasonably expected to vary mainly via sinks (i.e., wet scavenging).

Lines 78-92: In Chemical transport models, time splitting approach is used for calculating each physical and chemical process (emission, transport/diffusion, chemical transformation, deposition (in-cloud nucleation, belowcloud scavenging, and dry deposition)). Each process needs to be calculated or parametrized as accurate as possible. To improve the parameterization of below-cloud aerosol scavenging, collecting field data through measuring aerosol concentrations before and after precipitation events covering different precipitation types and intensities would be the best approach, in my opinion. The approach used in this study involves both in-cloud and below-cloud scavenging contributions as well as additional entrainment of pollutants along the air mass trajectory, and is not possible to quantify precipitation scavenging. I have difficulties in finding the true scientific value in such an analysis for improving our understanding in below-cloud aerosol scavenging (which seems to the major goal of this study).

Response: As noted in the Limitations section (Sect. 4), this method cannot isolate specific processes such as in-cloud or below-cloud scavenging. Because of this limitation, the objective of our study is not solely below-cloud scavenging but to identify potential indicators for aerosol scavenging (including both in- and below-cloud scavenging). The novelty of the study includes the following:

- It uses aircraft field campaign data collected over the West Pacific that hosts a wide range of transport patterns, aerosol sources, and cloud-precipitation systems. This presents the opportunity to use this aircraft data to study scavenging during long-range transport, which is currently a large uncertainty in chemical transport models.

- Since we found that APT inaccurately predicts $TE_{BC}$, our study recommends alternative meteorological variables that future scavenging studies can use instead ($RH_{q90}$, $f_{RH95}$, $f_{MR15}$, and PI). This information is important when high-resolution aerosol chemical transport modeling is not feasible/accessible for relating aerosol scavenging to meteorology. Furthermore, this suggests the possibility that models should also consider RH in addition to precipitation when estimating scavenging.

- The method we present in this study can be applied on multiple environments (e.g., multiple field campaign datasets) to identify potential scavenging indicators that could vary regionally.

Line 11 and line 31: This sentence implies that wet scavenging is the only dominant sink for aerosol particles. I would consider both dry deposition and wet scavenging as dominant sinks. Maybe change to "As one of the dominant sinks".

Response: The phrasing has been updated.

Lines 43-44: You should also include "precipitation intensity, amount, frequency, type". Precipitation type includes liquid and solid (snow) precipitation.

Response: The updated sentence now reads "The efficiency of below-cloud scavenging depends on raindrop size distributions, aerosol composition (Lu and Fung, 2018; Grythe et al., 2017), the amount of in-cloud condensed water (Luo et al., 2019) as well as precipitation characteristics (i.e., frequency, intensity, amount, and type)."

Lines 41-50: I would like to draw your attention of a series of studies on below-cloud aerosol scavenging conducted by a group in Environment Canada to enhance the discuss presented in this paragraph. They not only systematically assessed important parameters affecting below-cloud aerosol scavenging (Wang et al., 2010, ACP, 10, 5685-5705; Wang et al., 2011, ACP 11, 11859-11866; Zhang et al., 2013, ACP 13, 10005-10025), but also developed a new set of semi-empirical parameterizations (Wang et al., 2014a, GMD, 7, 799–819; 2014b, JAMES, 6, 1301-1310).

Response: Thank you for the references. We have added these studies to the discussion in the Introduction, which

now reads:

"Wang et al. (2010) determined the below-cloud scavenging coefficient is influenced by (1) raindrop-particle collection efficiency, (2) raindrop size distribution, and (3) raindrop terminal velocity. These factors were associated with differences in particle concentrations by a factor of 2 for sub-10 nm particles and a factor of >10 for particles larger than 3 µm; however, their combined uncertainty was insufficient to explain the discrepancy between theoretical and field measurements of the below-cloud scavenging coefficient. Wang et al. (2011) demonstrated that this discrepancy can be largely explained by the vertical turbulence as it determines which particles are subjected to impaction scavenging. This impact was most pronounced for submicron particles under weak precipitation intensities.

Given these uncertainties, Wang et al. (2014a) developed a new semi-empirical, size-resolved parametrization based on an percentile-logarithmic power-law relationship between the below-cloud scavenging coefficient and particle size that is applicable to both rain and snow across different particle sizes and precipitation intensities. Based on the size-resolved parametrization of Wang et al. (2014a), a bulk or modal parametrization for fine ($PM_{2.5}$), coarse ($PM_{2.5-10}$), and giant particles ($PM_{10+}$) was presented by Wang et al. (2014b)."

Line 97: Was the experiment cover one single site or one big area?

Response: Flights were conducted within a domain spanning 5 – 20°N, 117 – 127°E. The text has been updated to reflect this information.

Lines 229 and below: RH is identified as a key variable, but it is likely because its directly link with precipitation. In this case, there is no need to include this additional variable in precipitation scavenging parameterization. If the effect of the high RH is through hydroscopic growth of particles, then this needs to be mentioned.

Response: While RH is indeed related to precipitation, the poorer performance of APT (lower correlation, greatly underestimates $TE_{BC}$, higher intercept/bias) compared to RH variables such as $RH_{q90}$ indicates that data products that are used to calculate APT (i.e., satellite retrievals or reanalysis) may be missing instances of precipitation where scavenging is happening. The ability of $RH_{q90}$ to predict $TE_{BC}$ suggests that perhaps these instances of missed precipitation appear to satellite or reanalysis products as areas of high humidity. This is a possibility that could be explored in future work.

**Citation**: https://doi.org/10.5194/egusphere-2023-726-RC2